# Transfer Learning for Bayesian Optimization on Heterogeneous Search Spaces

**Zhou Fan**                                                                    *zfan@g.harvard.edu*
*Harvard University*

**Xinran Han**                                                                  *xinranhan@g.harvard.edu*
*Harvard University*

**Zi Wang**                                                                     *wangzi@google.com*
*Google DeepMind*

**Reviewed on OpenReview:** *https://openreview.net/forum?id=emXh4M7TyH*

## Abstract

Bayesian optimization (BO) is a popular black-box function optimization method, which makes sequential decisions based on a Bayesian model, typically a Gaussian process (GP), of the function. To ensure the quality of the model, transfer learning approaches have been developed to automatically design GP priors by learning from observations on "training" functions. These training functions are typically required to have the same domain as the "test" function (black-box function to be optimized). In this paper, we introduce MPHD, a *model pre-training* method on *heterogeneous domains*, which uses a neural net mapping from domain-specific contexts to specifications of hierarchical GPs. MPHD can be seamlessly integrated with BO to transfer knowledge across heterogeneous search spaces. Our theoretical and empirical results demonstrate the validity of MPHD and its superior performance on challenging black-box function optimization tasks.

## 1 Introduction

Many real-world applications require finding the best hyperparameter values by evaluating a series of configurations of those hyperparameters. Some examples include tuning machine learning (ML) models (Snoek et al., 2012; Turner et al., 2021), learning robot control strategies (Wang et al., 2021), synthesizing functional chemicals (Shields et al., 2021), and discovering new materials (Frazier & Wang, 2015). For these problems, there exists an underlying black-box function that scores the utility of hyperparameters. One popular way of formulating such problems is Bayesian optimization (BO): optimizing an unknown function by reasoning about the Bayesian beliefs about this function.

In BO, we often use Gaussian processes (GPs) as Bayesian beliefs for unknown functions. Given a GP prior and observations on the function, we can obtain a GP posterior and use its uncertainty predictions to make decisions on which datapoints to acquire. For example, one popular strategy is to greedily evaluate the inputs that achieve the highest upper confidence bounds on function values (Srinivas et al., 2010).

A prerequisite of BO is to specify a GP prior, which can be difficult to do in practice. To address this issue, much progress has been made to learn the GP prior using transfer learning based approaches (Swersky et al., 2013; Yogatama & Mann, 2014; Wang et al., 2018; Perrone et al., 2018; Volpp et al., 2019; Wistuba & Grabocka, 2021; Wang et al., 2022). These approaches typically assume that we have data on a set of "training" functions, and the goal is to generalize a learned GP model or a learned strategy to a "test" function, which has the *same domain* as the training functions. While these methods have been shown to perform well on a variety of tasks, they cannot be easily used to generalize to test functions that do not share the same domain as the training functions.

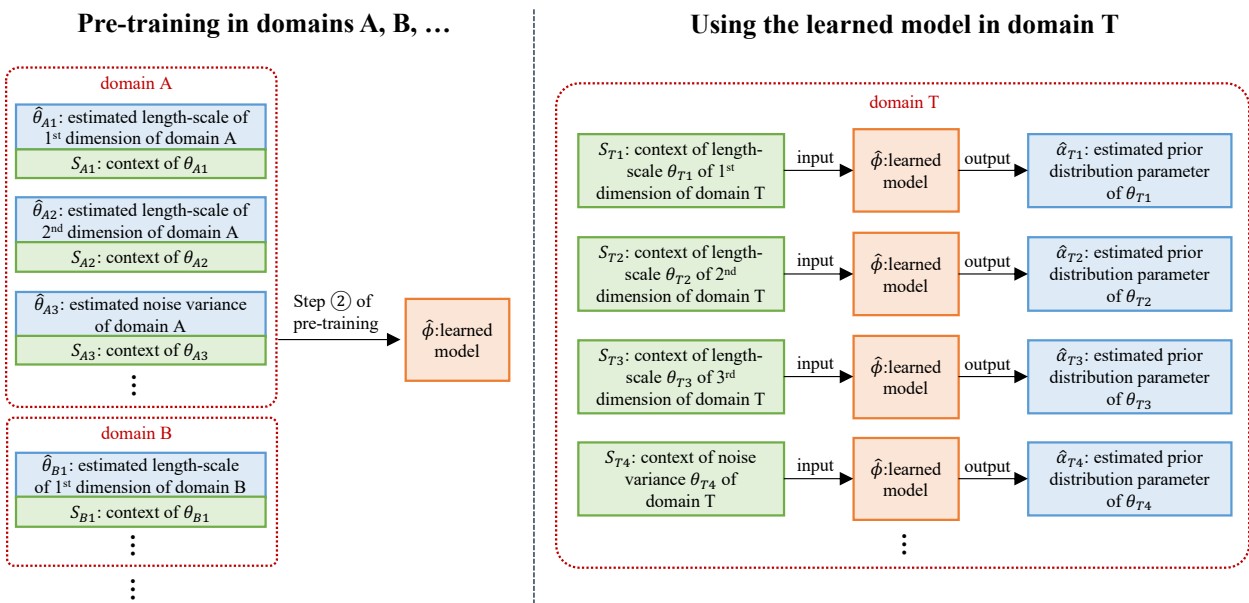

Figure 1: Illustration of how the learned MPHD model is used for BO. Only the second step of the pre-training of MPHD is shown. In this example, domain A has 2 dimensions so there are 2 length-scale parameters for it, while domain T has 3 dimensions and there are 3 corresponding length-scale parameters.

In practice, the data available for transfer learning might not have the ideal setups to ensure the domains of all functions are well-aligned. For example, we might have collected many datapoints by experimenting with several robot skills that have different (but potentially overlapping) sets of control parameters (Wang et al., 2021). Or, we might have ML model tuning data from a commercial BO tool (Golovin et al., 2017; Balandat et al., 2020), where different people might tune different hyperparameters and use different names for the same type of hyperparameters. With current methods, it can be difficult to transfer knowledge from one task to another in these cases.

In this paper, we introduce Model Pre-training on Heterogeneous Domains (MPHD), a new transfer learning method for BO on heterogeneous search spaces. In the pre-training stage, MPHD learns a model with a mapping from domain-specific contexts to specifications of hierarchical GPs. This allows transferring knowledge from a set of training functions with different domains to a new test function to be optimized on an unseen search space. Using the pre-trained model, MPHD can generate a customized hierarchical GP as the prior for the test function, and then this hierarchical GP can be combined with an existing acquisition strategy to perform BO. An illustration can be found in Fig. 1.

Through theoretical and empirical case studies (§3), we show that MPHD is asymptotically consistent, meaning that it converges to the ground truth solution as the number of training functions increases. We also show that the hierarchical GPs generated by MPHD can accurately capture test functions with new domains.

To verify the usefulness of MPHD for BO (§4), we conducted extensive experiments on real world BO transfer learning problems with heterogeneous search spaces. We tested benchmarks including HPO-B (Pineda-Arango et al., 2021) and PD1 (Wang et al., 2022), which involve 17 search spaces in total. Our results have shown significant improvement made by MPHD on sample efficiency for BO on functions with unseen search spaces.

In this paper, we make three **contributions**: (1) We identify a practical problem in BO, and propose a new problem formulation: the transfer learning problem for functions with different domains. (2) We propose a new method, MPHD, to solve this problem. (3) We show the effectiveness of MPHD both theoretically and experimentally, and prove the consistency of MPHD building on our new theoretical results for constructing sufficient statistics of training functions. To the best of our knowledge, MPHD is the first GP-based framework that can be used to transfer knowledge for BO on heterogeneous search spaces.

**Related work.** Researchers have developed methods for transferring knowledge between BO tasks. For example, Swersky et al. (2013) and Yogatama & Mann (2014) proposed to learn a multi-task GP and use the similarity between tasks for generalization. Feurer et al. (2018) aimed to learn warm-starting strategies from previous BO tasks using an ensemble. Recently, Wang et al. (2018); Perrone et al. (2018); Wistuba & Grabocka (2021); Wang et al. (2022) found that learning a GP prior from evaluations on training functions can be an effective approach to transfer knowledge for BO if all the functions share the same domain.

Another line of related work is end-to-end black-box function optimization. Chen et al. (2017) trained a recurrent neural network (RNN) on a large number of BO trajectories. The RNN can then be used to generate the next point to evaluate for a new BO problem. Chen et al. (2022) introduced OptFormer, a transformer-based transfer learning method for hyperparameter tuning on universal search spaces. OptFormer trains a giant transformer model on millions of BO trajectories to learn to propose hyperparameters in an end-to-end fashion. Note that Feurer et al. (2018) and Chen et al. (2017; 2022) all require previous optimization runs, meaning that they cannot make use of raw evaluations on training functions without simulating BO trajectories. Our approach, MPHD, focuses on transferring knowledge about functions by pre-training a surrogate model on raw evaluations, and does not require data in the form of BO trajectories.

MPHD can be naturally combined with other components of BO methods, e.g. acquisition functions, input and output warping, cross validation etc, to complete a practical BO software. For example, MPHD can be directly incorporated in BoTorch (Balandat et al., 2020) and Vizier (Golovin et al., 2017) by replacing their default hierarchical GP model.

## 2 MPHD: Model Pre-training in Heterogeneous Domains

We present MPHD, a model pre-training framework for functions with heterogeneous domains. Given data collected on training functions, MPHD aims to learn a distribution over functions to model an unseen test function. The domains of all training functions and the test function can have different numbers of dimensions and each input dimension can have different meanings.

### 2.1 Problem formulation

We first define terms on datasets. A *super-dataset* is a collection of data points from all training functions with different domains, along with the contexts associated with each domain. A *dataset* is a collection of data points from training functions with the same domain. A *sub-dataset* is a collection of data points from the same training function. Fig. 3 illustrates a super-dataset example.

Formally, we use $D = \{(D_i, S_i)\}_{i=1}^N$ to denote a super-dataset, where $D_i$ is a dataset and $S_i$ is a domain-specific context. Each dataset $D_i$ consists of observations on a collection of training functions $F_i = \{f_{ij} : \mathcal{X}^{(i)} \to \mathbb{R}\}_{j=1}^{M_i}$ where functions in $F_i$ share the same compact domain $\mathcal{X}^{(i)} \subset \mathbb{R}^{d_i}$. The information about domain $\mathcal{X}^{(i)}$ is encoded into the context $S_i$. Let dataset $D_i = \{D_{ij}\}_{j=1}^{M_i}$, where each sub-dataset $D_{ij} = \{(x_{ij}^{(l)}, y_{ij}^{(l)})\}_{l=1}^{L_{ij}}$. $L_{ij}$ is the number of observations on function $f_{ij}$ perturbed by *i.i.d.* additive Gaussian noise, i.e., $y_{ij}^{(l)} \sim \mathcal{N}(f_{ij}(x_{ij}^{(l)}), \sigma_i^2)$. Noise variance $\sigma_i^2$ is specific to each domain $\mathcal{X}^{(i)}$.

We assume that all functions in $F_i$ with domain $\mathcal{X}^{(i)}$ are *i.i.d.* function samples from the same Gaussian process, $\mathcal{GP}_i = \mathcal{GP}(\mu_i, k_i; \theta_i)$, where $\mu_i$ is a constant mean function, $k_i$ is a stationary kernel, and $\theta_i = [\theta_{ih}]_{h=1}^{H_i} \in \mathbb{R}^{H_i}$ are GP parameters[1], including parameters of the mean and kernel functions as well as the noise variance $\sigma_i^2$. Each GP parameter $\theta_{ih}$ is sampled independently from its prior distribution $\Theta(\alpha_{ih})$, where $\alpha_{ih} = \phi(s_{ih}) \in \mathbb{R}^{d_\alpha}$, $s_{ih}$ is the context of GP parameter $\theta_{ih}$, and $\phi$ maps from contexts to hyperparameters (parameters of the priors). The domain-specific context $S_i$ is composed of the contexts of all GP parameters, i.e., $S_i = [s_{ih}]_{h=1}^{H_i}$.

**Example of the domain-specific context.** For a $d$-dimensional domain using the Matern kernel, there are $d$ length-scale parameters. In the upcoming experiments in §4, each length-scale parameter is associated

---

[1]In the classic GP literature (Rasmussen & Williams, 2006), $\theta_i$ are called hyperparameters of a GP. But from a functional perspective, a GP is a parameterized distribution over random functions, and so $\theta_i$ can also been seen as parameters of a GP.

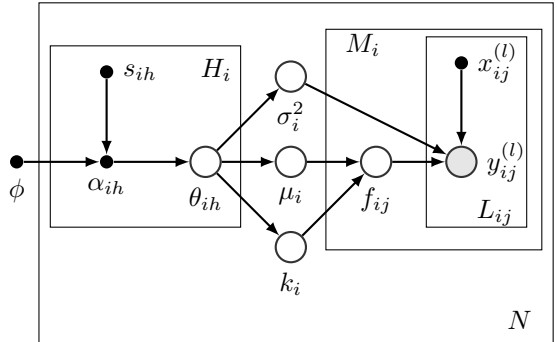
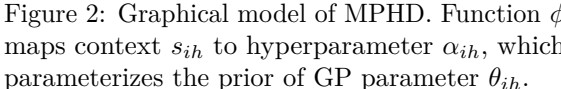

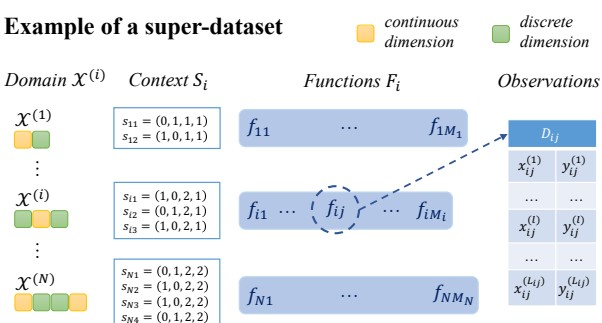

Figure 2: Graphical model of MPHD. Function $\phi$ maps context $s_{ih}$ to hyperparameter $\alpha_{ih}$, which parameterizes the prior of GP parameter $\theta_{ih}$.

Figure 3: An example of a super-dataset structure. Each dimension of domain $\mathcal{X}^{(i)}$ corresponds to either a discrete or continuous real-valued variable.

with a 4-dimensional context vector. This context vector comprises one-hot encoding for the first two dimensions, identifying the domain dimension that the length-scale parameter corresponds to as either discrete or continuous. The remaining two dimensions of the context vector denote the counts of discrete and continuous dimensions within the domain respectively. For example, the domain marked as $\mathcal{X}^{(i)}$ in Fig. 3 includes 2 discrete dimensions (first and third) and 1 continuous dimension (second). Consequently, the context vector for the length-scale parameter of its first domain dimension is expressed as $(1, 0, 2, 1)$, indicating a discrete dimension in a domain having two discrete dimensions and one continuous dimension. The context vector for the second domain dimension is represented as $(0, 1, 2, 1)$. One advantage of this basic context representation is that it can be automatically generated for almost any domains, requiring only the essential meta description of domain for BO.

Experiment results in §4 will showcase that this basic domain-specific context, although simple, is surprisingly effective in conveying information about prior distributions in our model. Intuitively, the prior over a continuous learning rate hyperparameter should be very different from the prior over a discrete hyperparameter which refers to an activation function type. Moreover, tuning problems on similar models tend to have a similar number of hyperparameters to tune. For example, in HPO-B (Pineda-Arango et al., 2021), the "svm" search spaces have 6 to 7 dimensions, while the "rpart" search spaces have 29 to 31 dimensions. There could be other ways to include more sophisticated information about the black-box function, and MPHD is a general method that is capable of incorporating any kinds of contexts in a vector format.

Fig. 2 illustrates the graphical model. The goal of MPHD is to pre-train this probabilistic model so that for any new domain, the model can generate the prior distributions over the GP parameters to construct a domain-specific hierarchical GP.

## 2.2 Our method

As shown in Fig. 2, the model can be compactly described by function $\phi$. Thus, model pre-training in MPHD is equivalent to training function $\phi$ on the super-dataset, $D = \{(D_i, S_i)\}_{i=1}^{N}$, such that the model can generalize to test functions with new contexts. We define the pre-training objective to be the log marginal data likelihood as follows,

$$\mathcal{L}(\phi) = \sum_{i=1}^{N} \log p(D_i \mid \phi, S_i) = \sum_{i=1}^{N} \log \int_{\theta_i} p(D_i \mid \theta_i) p(\theta_i \mid \phi, S_i) \, d\theta_i$$

$$\approx \sum_{i=1}^{N} \log \sqrt{\frac{(2\pi)^{H_i}}{M_i^{H_i} \det \frac{d^2}{d\theta^2}(-\frac{1}{M_i} \log p(D_i \mid \theta_i))|_{\theta_i = \hat{\theta}_i}}} p(\hat{\theta}_i \mid \phi, S_i) p(D_i \mid \hat{\theta}_i) \tag{1}$$

$$\propto \sum_{i=1}^{N} \left( \log p(D_i \mid \hat{\theta}_i) + \log p(\hat{\theta}_i \mid \phi, S_i) \right) \propto \sum_{i=1}^{N} \log p(\hat{\theta}_i \mid \phi, S_i) \tag{2}$$

where $\hat{\theta}_i = \arg\max_{\theta_i} p(D_i \mid \theta_i)$ and $\theta_i = [\theta_{ih}]_{h=1}^{H_i}$. The approximation in Eq. 1 uses Laplace's method (see derivations in §B). Eq. 2 removes terms irrelevant of optimizing $\phi$. To summarize, model pre-training in MPHD has two steps:

$$\text{①} \ \forall i \in [N], \hat{\theta}_i \leftarrow \arg\max_{\theta_i} p(D_i \mid \theta_i), \ \ \text{②} \ \hat{\phi} \leftarrow \arg\max_{\phi} \sum_{i=1}^{N} \log p(\hat{\theta}_i \mid \phi, S_i).$$

Step ① can be done using gradient-based optimization methods for each of the $N$ likelihood functions,

$$\log p(D_i \mid \theta_i) = \sum_{j=1}^{M_i} \log p(D_{ij} \mid \theta_i) = -\sum_{j=1}^{M_i} \left( \left(\boldsymbol{y}_{ij}^{(\theta_i)}\right)^{\top} \left(K_{ij}^{(\theta_i)}\right)^{-1} \boldsymbol{y}_{ij}^{(\theta_i)} + \frac{1}{2} \log |K_{ij}^{(\theta_i)}| + \frac{L_{ij}}{2} \log 2\pi \right), \quad (3)$$

where vector $\boldsymbol{y}_{ij}^{(\theta_i)} = [(y_{ij}^{(l)} - \mu_i(x_{ij}^{(l)}; \theta_i)]_{l=1}^{L_{ij}}$ and matrix $K_{ij}^{(\theta_i)} = [k_i(x_{ij}^{(l)}, x_{ij}^{(l')}; \theta_i)]_{l=1, l'=1}^{L_{ij}}$.

Step ② requires computing the approximated objective

$$\mathcal{L}(\phi) \approx \hat{\mathcal{L}}(\phi) = \sum_{i=1}^{N} \log p(\hat{\theta}_i \mid \phi, S_i) = \sum_{i=1}^{N} \sum_{h=1}^{H_i} \log p_{\Theta}(\hat{\theta}_{ih} \mid \alpha_{ih} = \phi(s_{ih})), \quad (4)$$

which depends on the exact forms of the prior distributions $\Theta(\alpha_{ih})$. Moreover, we need to specify the function space for $\phi$ such that optimization is possible. In this work, we use a neural network to parameterize function $\phi$, and Step ② can be done by optimizing over the weights in $\phi$ with gradient-based methods.

## 3 Case studies for validating MPHD

We validate MPHD via case studies. §3.1 presents theoretical analyses on convergence and consistency. §3.2 shows empirical results on synthetic data to compare pre-trained models with the ground truth.

### 3.1 Theoretical analyses

For the theoretical analysis in this section, we assume zero mean and anisotropic Matérn kernels with known smoothness term $\nu$ (e.g. $\nu = 5/2$), and the GP parameters $\theta_i = [\theta_{ih}]_{h=1}^{H_i}$ to be learned only include length-scales. For simplicity, we also assume that contexts $s_{ih}$ are the same across domains $\mathcal{X}^{(i)}, i \in [N]$ and the length-scale priors are normal or Gamma distributions. Under mild conditions, we show that (1) For each $i \in [N]$, as $M_i$ (the number of sub-datasets) increases, the estimated GP parameters $\hat{\theta}_i$ for the respective domain $\mathcal{X}^{(i)}$ converge to the ground truth parameters $\theta_i^*$, and consequently (2) For each $h \in [H_i]$, as $N$ (the number of datasets) increases, the learned hyperparameters $\hat{\alpha}_{ih}$ of the normal or Gamma priors converge to the ground truth values $\alpha_{ih}^*$.

**Background.** The theoretical soundness of pre-training on heterogeneous domains relies on the quality of the estimated GP parameters from each domain. The asymptotic behavior of MLE for covariance parameters under a single GP has commonly been studied in two asymptotic frameworks with fixed or increasing domains (Bevilacqua et al., 2019; Zhang & Zimmerman, 2005). Under the *fixed domain* setting, observations are sampled in a bounded set and thus become increasingly dense with more samples. In the *increasing domain* setting, observations are collected with a minimal spacing in between data points and thus make the sampling domain unbounded as the number of observations increases; i.e., for a sub-dataset $\{(x^{(t)}, y^{(t)})\}_{t=1}^{T}$ with $T$ observations, there exists a fixed $\Delta > 0$ such that

$$\|x^{(t)} - x^{(t')}\| \geq \Delta, \quad \forall t \neq t', 1 \leq t, t' \leq T. \quad (5)$$

Mardia & Marshall (1984) and Stein (1999) showed that, given the observations from a single function sampled from a zero-mean GP, MLE estimates of covariance parameters are consistent and asymptotically normal with mild regularity conditions for increasing domains. Bachoc (2014) showed that while the MLE given finite observations may not be unique, it converges to a unique global minimizer with probability converging to one as $T$ goes to infinity. Additionally, Bachoc et al. (2020) discusses extensions of these results to GP models with non-zero mean functions.

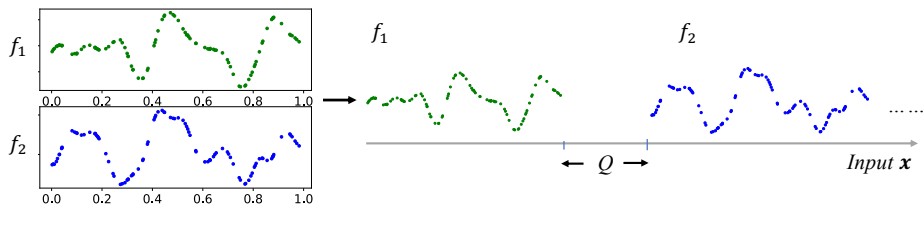

Figure 4: Illustration of how to construct a pseudo sub-dataset in Lemma 1 from observations on functions $f_1, f_2, \cdots$. We rearrange the original sub-datasets in the domain such that datapoints from different sub-datasets have a distance of at least $Q > 0$ in the constructed pseudo sub-dataset.

**Our results.** The critical part of our proof is to show that the setup of MPHD belongs to the increasing domain setting. The key property of the increasing domain setting is that there is vanishing dependence between observations that are far apart (Bachoc, 2014). Thus, a larger sample size would be more informative of the covariance structure.

**Lemma 1.** *For any $i \in [N]$, $\mathcal{GP}_i(\mu_i, k_i; \theta_i)$ and its corresponding* dataset $D_i = \{D_{ij}\}_{j=1}^{M_i}$, *there exists a pseudo sub-dataset $\bar{D}$ observed at inputs $x^{(1)}, x^{(2)}, \ldots, x^{(T)}$ on a function $f' \sim \mathcal{GP}_i$, such that $\bar{D}$ satisfies the increasing domain assumption in Eq. 5, and $\bar{D}$ is a sufficient statistic of $D_i$ with $p(D_i \mid \theta_i) \equiv p(\bar{D} \mid \theta_i)$.*

*Proof.* Consider functions $f_{ij}$ whose observations $D_{ij} = \{(x_{ij}^{(l)}, y_{ij}^{(l)})\}_{l=1}^{L_{ij}}$ is in dataset $D_i = \{D_{ij}\}_{j=1}^{M_i}$, where $j \in \{1, \ldots, M_i\}$.

Since the domain $\mathcal{X}^{(i)}$ of $f_{ij}$ is compact, and we define $Q' = \sup_{x,x' \in \mathcal{X}^{(i)}} \|x' - x\| + 1 \ll \infty$. Since the stationary covariance function only concerns the relative location between two points, we can shift all inputs jointly while the NLL, $\text{NLL}(\{D_{ij}\}) = -\log p(\{D_{ij}\} \mid \theta_i)$, stays the same. More formally, let $\bar{D}_j = \{(x_{ij}^{(l)} + \mathbf{1}_{d_i}(Q + Q')j, y_{ij}^{(l)})\}_{l=1}^{L_{ij}}$ for some $Q > 0$, where $d_i$ is the input dimension and $\mathbf{1}_{d_i}$ is a $d_i$−dimensional vector filled with ones. Because the distance between any pairs of inputs stays the same, we have $\text{NLL}(\{D_{ij}\}) = \text{NLL}(\{\bar{D}_j\}) = -\log p(D_{ij} \mid \theta_o)$.

We define an augmented sub-dataset $\bar{D} = \bar{D}_1 \cup \cdots \cup \bar{D}_{M_i}$.

As $Q \to \infty$, we can show that the sum of NLLs over a set of sub-datasets and the NLL of the single augmented sub-dataset are equivalent. Without loss of generality, consider the Matérn covariance function with smoothness term $\nu = 1/2$ and variance $\sigma^2$, length-scale $\rho$: $C_{1/2}(d) = \sigma^2 \exp(-\frac{d}{\rho})$. Since the covariance exponentially decays with the distance $d$ between points, for any $\epsilon > 0$, we can find a $Q$ such that $C(d) < \epsilon$ for $d > Q$.

For the augmented sub-dataset $\bar{D} = \bar{D}_1 \cup \cdots \cup \bar{D}_{M_i}$, we have $\|x' - x\| > Q$ for any $x \in \bar{D}_j$ and $x' \in \bar{D}_{j'}$, $j \neq j'$. As $\epsilon \to 0$, the covariance matrix for this augmented sub-dataset becomes block diagonal. As a result, we can express its NLL as a sum of NLL for $D_{i1}, \ldots, D_{iM_i}$ respectively, i.e., $\text{NLL}(\{\bar{D}\}) = -\sum_{j=1}^{M_i} \log p(\bar{D}_j \mid \theta_i)$. This gives us the equivalence of density in Lemma 1, and by definition, $\bar{D}$ is a sufficient statistic of the original dataset $D_i$.

Note that the distance between inputs in $\bar{D}$ will always be at least $\min_j \min_{l \neq l'} \|x_{ij}^{(l)} - x_{ij}^{(l')}\| > \delta$, since any two datapoints in each sub-dataset $D_{ij}$ are at least $\delta$ apart by construction in problem setup. As $M_i \to \infty$, the augmented sub-dataset $\bar{D}$ is unbounded and satisfies the increasing domain characterization in Eq. 5. $\square$

Lemma 1 highlights the important connection between the increasing domain setting and our setups with an increasing number of sub-datasets. Intuitively, this lemma shows that observations from multiple independently generated functions from a fixed GP can be viewed as being sampled from one function, with infinitely large interval between sub-datasets' observations. We illustrate this process on a 1-dimensional domain in Fig. 4. We prove Lemma 1 in §C.

To show the asymptotic properties of MPHD, we make the following **assumptions**:

(1) Dataset $D_i = \{D_{ij}\}_{j=1}^{M_i}$ ($i \in [N]$) contains a finite number of observations in each of its sub-datasets. There exists a minimum spacing $\delta > 0$ such that $\|x_{ij}^{(l)} - x_{ij}^{(l')}\| \geq \delta, (l \neq l')$ for all sub-datasets $D_{ij}$.

(2) The ground truth GP parameters $\theta_i^*$ belong to $C_i$, the space of $\theta_i$, used for estimating $\hat{\theta}_i$ in Step ①.

(3) For each $i \in [N]$, there is sufficient information in sampling locations $x^{(1)}, x^{(2)}, \ldots, x^{(T)}$ of the pseudo sub-dataset $\tilde{D}$ (Lemma 1) to distinguish covariance functions $k_i(\cdot, \cdot; \theta_i)$ with $k_i(\cdot, \cdot; \theta_i^*)$ (Bachoc, 2014). That is, for any $\theta_i \in C_i$,

$$\liminf_{T \to \infty} \inf_{\|\theta_i - \theta_i^*\| \geq \epsilon} \frac{1}{T} \sum_{t,t'=1}^{T} \left( k_i(x^{(t)}, x^{(t')}; \theta_i) - k_i(x^{(t)}, x^{(t')}; \theta_i^*) \right)^2 > 0. \qquad \text{(Asymptotic Identifiability)}$$

Intuitively, the asymptotic identifiability condition means that no two distinct covariance parameters exist such that the two covariance functions are the same on the set of randomly sampled points. In our case, as the number of observations $T \to \infty$ in an unbounded region (increasing domain), it is realistic to expect that this condition holds on the sec of observations.

**Theorem 2.** *Given assumptions (1)-(3), for dataset $D_i$ with $M_i$ sub-datasets generated from the same mean and covariance function, as $M_i \to \infty$, we have $\hat{\theta}_i \xrightarrow{p} \theta_i^*$.*

**Theorem 3.** *Given assumptions (1)-(3), as the number of datasets, and sub-datasets $N, M_i \to \infty, \forall i$, MLE for the prior distribution of each GP parameter $\Theta(\alpha_{ih}), h \in [H_i]$ is consistent; i.e., $\hat{\alpha}_{ih} \xrightarrow{p} \alpha_{ih}^*$.*

The proofs of Theorem 2 and 3 can be found in §C. The above theorems complete the claims we made at the beginning of §3.1. Moreover, MPHD has the following asymptotic MLE behaviors in Step ② with $n$ estimated samples of GP parameters $\hat{\theta}_i$.

**Remark.** *(Ye & Chen (2017) and Rice (2006))*
*(1) When the prior is assumed to be from a Gamma distribution parameterized by shape $a^*$ and rate $b^*$, the MLE $(\hat{a}, \hat{b})$ is consistent and asymptotically normal with distribution $\mathcal{N}((a^*, b^*), \frac{1}{n}\mathcal{I}(a^*, b^*)^{-1})$, where $\mathcal{I}$ is the Fisher information matrix.*
*(2) Similar results hold when the prior is a normal distribution parametrized by mean $c^*$ and standard deviation $d^*$, where the MLE satisfies $(\hat{c}, \hat{d}) \to (c^*, d^*)$.*

A key observation from our theoretical analyses is that with sufficient observations in each dataset $D_i$, increasing the number of sub-datasets can effectively improve the estimation of the covariance parameters and thus the parameters of the prior distribution. We show the empirical asymptotic behavior in §3.2.

## 3.2 Empirical analysis with synthetic data

In this section, we present empirical results to further demonstrate the asymptotic properties of MPHD. For these empirical results, we assume constant mean and anisotropic Matérn kernels with known smoothness term $\nu$. The asymptotic properties of learning the length-scale parameter are included in this section, while the results for other GP parameters can be found in §F. We generated two synthetic super-datasets following the generative process illustrated in Fig. 2 with two different settings where we use Gamma distributions as the prior for length-scales.

**Synthetic Super-dataset (S)** is a smaller and simpler one out of the two synthetic super-datasets, which uses the same Gamma prior for all domains (i.e., function $\phi$ returns constant hyperparameters). It includes 20 datasets (each with its own domain) with 10 sub-datasets in each dataset. Each sub-dataset includes noisy observations at 300 random input locations in its respective domain. The dimension of each domain is randomly sampled between 2 and 5.

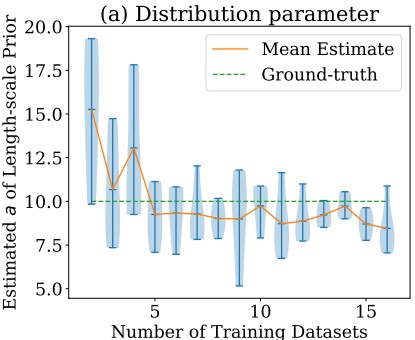 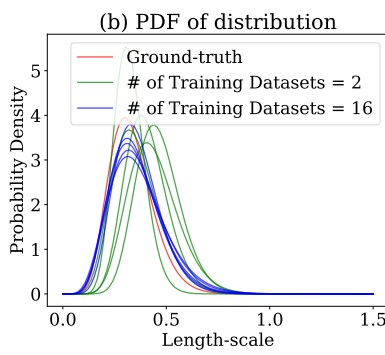 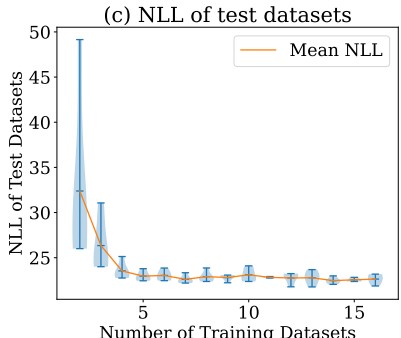

Figure 5: For Synthetic Super-dataset (S) with a fixed one-dimensional length-scale prior, we plot (a) estimated shape parameter $\hat{a}$ of Gamma distribution prior for the length-scale GP parameter, (b) the PDF of the shared Gamma prior for length-scales (§3.2.1), and (c) NLLs of the test datasets on pre-trained priors w.r.t. the number of training datasets. We show the mean and violin plots over 5 random seeds. The pre-trained Gamma distributions with 16 training datasets are more stable than those with 2 training datasets and match well with the ground-truth prior.

**Synthetic Super-dataset (L)** is a larger and more complex super-dataset and is also the synthetic data used for BO evaluations in §4. It has domain-specific Gamma priors where the hyperparameters linearly depend on the number of dimensions of each domain (i.e., function $\phi$ is a linear function of the domain dimension $d_i$). We let the hyperparameters depend on the number of input dimensions in order to simulate practical applications where tuning objectives with different numbers of input dimensions might need to be modeled with very different hyperparameters in MPHD. Additionally, this synthetic linear dependency allows for the comparison of learned hyperparameters against the actual ground truth. This super-dataset includes 20 datasets (each with its own domain) with 20 sub-datasets in each dataset. Each sub-dataset includes noisy observations at 3000 random input locations in its respective domain. The dimension of each domain is randomly sampled between 2 and 14.

We split each of the synthetic super-datasets into training data and test data: for Synthetic Super-dataset (S), we used 80% of datasets as training datasets and the remaining 20% as test datasets; for Synthetic Super-dataset (L), we used 80% sub-datasets within each dataset as the training sub-datasets and the remaining 20% as the test sub-datasets. All experiments are repeated 5 times with different random seeds. More details on the setups of the synthetic super-datasets can be found in §E.1.

### 3.2.1 Length-scales with the same Gamma prior

For Synthetic Super-dataset (S), Fig. 5(a) shows $\hat{a}$ of the pre-trained Gamma prior $\Gamma(\hat{a}, \hat{b})$ for length-scales, where $\hat{a}$ is the shape parameter and $\hat{b}$ is the rate parameter. As the number of training datasets increases, the variance of the estimated $\hat{a}$ gradually decreases and the mean becomes closer to the ground-truth prior. Fig. 5(b) plots the PDF of both the pre-trained and ground-truth Gamma priors, showing that more training datasets help the stability of pre-training. These results are consistent with our theoretical analyses in §3.1.

In Fig. 5(c), we show the NLL of test datasets (the negative of the objective $\mathcal{L}(\phi)$ in §2.2, but applied on test data instead of training data) with an increasing number of training datasets. Both the mean and variance of the NLL drop as the number of training datasets increases, indicating no overfitting.

### 3.2.2 Length-scales with domain-specific Gamma priors

For Synthetic Super-dataset (L) with domain-specific Gamma length-scales priors, we ran MPHD with two versions of the function $\phi$.

**Variants of MPHD:** (1) MPHD Standard, which uses a neural net (NN) to represent the length-scale prior for any domain dimension taking as input the context $s_{ih}$ that corresponds to the length-scale of that domain dimension and generating the length-scale Gamma prior for that domain dimension as output. On

Synthetic Super-dataset (L), MPHD Standard uses a domain-specific context that specifies the number of dimensions as all domain dimensions are continuous. For each of the other GP parameter types such as signal variance, MPHD Standard directly learns a shared prior distribution without NN for all domains. The NN-based length-scale prior of MPHD Standard consists of two hidden layers, each of size 16, and utilizes the hyperbolic tangent as the activation function. When this NN takes as input the context vector related to a length-scale parameter, it outputs a 2-dimensional vector. This output vector symbolizes the two hyperparameters (shape and rate) of the Gamma distribution that form the length-scale prior distribution. (2) MPHD Non-NN HGP: the model is a simplified version that learns a shared length-scale Gamma prior for all search space dimensions without using an NN-based length-scale prior and is pre-trained with the two-step approach. Essentially, function $\phi$ outputs a constant for every hyperparameter type, same as §3.2.1. Priors for the other GP parameter types such as signal variance are the same as MPHD Standard.

For both versions of MPHD, the underlying GP uses constant mean and anisotropic Matérn kernels, and function $\phi$ outputs hyperparameters in Gamma priors.

For the test datasets in Synthetic Super-dataset (L) with domain-specific length-scale priors, Fig. 6 presents the average KL divergence between the ground-truth Gamma priors for length-scales and the corresponding Gamma distributions generated by the pre-trained function $\hat{\phi}$, with an increasing number of training datasets. The results are presented for two versions of MPHD: the first empolying an NN-based prior and the second utilizing a non-NN prior. The KL divergence with respect to ground-truth Gamma prior is averaged over 5 repeats of experiments (with different random seeds) and all possible numbers of domain dimensions between 2 and 14. The KL divergence values of both versions of MPHD decrease as the number of training datasets increases, which shows that the pre-trained model gradually becomes closer to the ground truth. Moreover, MPHD with NN $\phi$ achieves lower KL divergence values than MPHD with constant $\phi$, showing the advantage of using an expressive function $\phi$ in MPHD for complex problems.

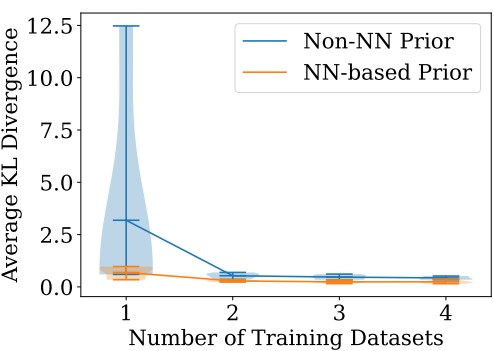

Figure 6: The average KL divergence between the ground-truth and the learned Gamma distributions.

## 4 MPHD for Bayesian optimization

The goal of BO is to optimize a black-box test function $f$ with as few evaluations on $f$ as possible. BO works by optimizing a series of acquisition functions to sequentially select inputs and observe their function values. In every iteration of BO, a surrogate model is constructed based on all the observations on function $f$, and the acquisition function is defined over the predictions from the surrogate model.

MPHD can generate domain-specific hierarchical GPs as surrogate models for BO on new search spaces. More precisely, to optimize function $f$ over its domain $\mathcal{X}^{(f)}$ (i.e., search space for BO[2]), we use the domain-specific context $S_f = [s_{fh}]_{h=1}^{H_f}$ of domain $\mathcal{X}^{(f)}$ to obtain the prior distributions $p(\theta_{fh} \mid \alpha_{fh})$, where $\alpha_{fh} = \phi(s_{fh}), \forall h \in [H_f]$. See Fig. 1 for an illustration of how MPHD generates the surrogate model.

At each iteration of BO, and we compute the MAP estimate for the GP parameters $\theta_f = [\theta_{fh}]_{h=1}^{H_f}$:

$$\hat{\theta}_f = \arg\max_{\theta_f} p(\theta_f \mid D_f, S_f, \phi = \hat{\phi}) = \arg\max_{\theta_f} p(D_f \mid \theta_f, S_f, \phi = \hat{\phi}) \prod_{h=1}^{H_f} p(\theta_{fh} \mid \alpha_{fh} = \hat{\phi}(s_{fh})), \quad (6)$$

where $D_f$ is the set of observations made on function $f$. We can then use a GP parameterized by $\hat{\theta}_f$, $\mathcal{GP}(\mu_f, k_f; \hat{\theta}_f)$, to compute the acquisition function. See the full algorithm in §D.

---

[2]Without loss of generality, we use "search spaces" and "domains" interchangeably in this paper.

In the rest of this section, we present the experiment results and analyses to verify the usefulness of MPHD for decision making in BO. §4.1 introduces the 3 types of datasets we experimented with. §4.2 lists the compared methods including 2 variants of MPHD and 9 different baselines. §4.3 presents the results on transfer learning for Bayesian optimization.

## 4.1 Datasets

We used both synthetic data and real-world data in our experiments. The synthetic data used was the Synthetic Super-dataset (L) introduced in §3.2.2, where the ground truth GP parameters for all domains were sampled from their prior distributions. The real-world data were collected on hyperparameter tuning tasks for classic machine learning models (Pineda-Arango et al., 2021) and near state-of-the-art deep learning models (Wang et al., 2022).

**HPO-B Super-dataset** (Pineda-Arango et al., 2021) is a large-scale multi-task benchmark for hyperparameter optimization. As a super-dataset, it consists of 16 different search spaces and more than 6 million evaluations in total. The dimensions of these search spaces vary from 2 to 18.

**PD1 Dataset** (Wang et al., 2022) is a large hyperparameter tuning dataset collected by training expensive deep neural network models on popular image and text datasets, as well as a protein sequence dataset. As a dataset (instead of a super-dataset), it contains 24 sub-datasets of the same 4-dimensional search space.

For HPO-B Super-dataset and PD1 Dataset, we normalized the range of every domain dimension as well as function values to $[0, 1]$. Synthetic Super-dataset (L) was generated with every domain dimension in $[0, 1]$, and its function values were kept unnormalized in order to test its ground-truth priors.

**Train/test splits:** For any super-dataset $D = \{(D_i, S_i)\}_{i=1}^N$ of the two, we split every dataset $D_i$ in the super-dataset into a training dataset $D_i^{\text{train}}$ and a test dataset $D_i^{\text{test}}$, each containing a disjoint subset of sub-datasets in $D_i$. As mentioned in §3.2.2, we used 80% sub-datasets within each dataset as the training sub-datasets and the remaining 20% as the test sub-datasets for Synthetic Super-dataset (L). HPO-B Super-dataset comes with a pre-specified per-dataset train/test split and we used the same setup.

To show the generalization capability across different real-world datasets, we evaluated the BO performances of MPHD on PD1 (Wang et al., 2022), where the model was pre-trained only on HPO-B (Pineda-Arango et al., 2021). Although MPHD models were never pre-trained on PD1 during the evaluation, a train/test split for PD1 is still needed because the homogeneous meta BO methods have to be pre-trained on PD1. Out of the 24 sub-datasets in PD1, we abandoned the ImageNet ResNet50 1024 task as it only has 100 datapoints. We randomly sampled 19 ($\sim 80\%$) of the remaining 23 sub-datasets as training sub-datasets and used the remaining 4 ($\sim 20\%$) sub-datasets as test sub-datasets. For convenience, we denote the entire PD1 Dataset by $D_P$, its training part by $D_P^{\text{train}}$, and its test part by $D_P^{\text{test}}$.

## 4.2 Experiment setups and compared methods

To test the capability of MPHD to generalize to new tasks with both seen and unseen search spaces, we designed experiments to compare MPHD with competitive meta BO baselines including HyperBO (Wang et al., 2022), which is known to be the state-of-the-art GP prior pre-training method for BO. For MPHD, we tested the performance of both the two variants MPHD Standard and MPHD Non-NN HGP to understand different ways of setting function $\phi$ in Fig. 2. As in the example shown in §2.1, MPHD Standard employs a 4-dimensional context vector for the length-scale parameter of each domain dimension. This context vector encodes information on whether the domain dimension is discrete or continuous, the number of discrete dimensions, and the number of discrete dimensions in the domain. The NN-based length-scale prior of MPHD Standard has the same architecture as in §3.2.2, with 2 hidden layers, each of size 16, and utilizes the hyperbolic tangent activation function.

**Baselines:** (1) Random sampling. (2) A Hand-specified Hierarchical GP prior, fixed across all search spaces. (3) A Non-informative Hierarchical GP prior, fixed across all search spaces. (4) HyperBO (Wang et al., 2022). (5) ABLR (Perrone et al., 2018). (6) FSBO (Wistuba & Grabocka, 2021). (7) Base GP, a single GP that uses the MLE of GP parameters $\hat{\theta}_i$ of the search space that it is being tested in, which is

the same MLE achieved in Step ① of the pre-training of MPHD. This baseline is essentially a simplified version of HyperBO that does not have the MLP base for its GP kernel and uses a constant mean function. (8) The Ground-truth Hierarchical GP prior, which is only available for Synthetic Super-dataset (L). (9) The Ground-truth GP for the test dataset, which is also only available for Synthetic Super-dataset (L).

HyperBO and FSBO use an anisotropic Matérn kernel ($\nu = 5/2$) with an MLP base, while ABLR uses a dot-product kernel with an MLP base. The sizes of layers in the MLP base are $(128, 128)$ for experiments on HPO-B Super-dataset and $(32, 32)$ on Synthetic Super-dataset. All the remaining GP-base methods use an anisotropic Matérn kernel ($\nu = 5/2$) without an MLP base. HyperBO uses a linear mean function with an MLP base. ABLR and FSBO use a zero mean function. The rest of the GP-based methods all use a constant mean function. Please see §E.2 for more details of the configuration of baselines, including GP prior parameters, of all compared methods.

Based on whether a method requires pre-training and how it is pre-trained, we can group all compared methods by the following categories: (1) Methods that are not pre-trained. This category includes Random, the Hand-specified HGP prior, the Non-informative HGP prior, the Ground-truth HGP prior, and the Ground-truth GP. (2) Heterogeneous meta BO methods, which are meta BO methods that can generalize over search spaces. This category only includes variants of MPHD. (3) Homogeneous meta BO methods, which are meta BO methods that can only train and test on a single search space. This category includes HyperBO, ABLR, FSBO, and Base GP.

When testing the BO performances of compared methods in the search space $\mathcal{X}^{(i)}$ corresponding to a dataset $D_i$ within a super-dataset $D$ (either Synthetic Super-dataset (L) or HPO-B Super-dataset), we used the test part of that dataset, $D_i^{\text{test}}$, to run BO with every tested method. For every method, we report its average normalized simple regret on all test sub-datasets in all search spaces, i.e., all sub-datasets in $\{D_i^{\text{test}}\}_{i=1}^N$. Methods that are not pre-trained can be directly tested. In order to test the ability of MPHD to generalize to new functions in seen search spaces as well as unseen search spaces, we designed two settings for its model pre-training. In the default setting, the MPHD models were pre-trained on training sub-datasets from all search spaces in the super-dataset, i.e., $\{(D_j^{\text{train}}, S_j)\}_{j=1}^N$. In the second setting denoted by NToT (Not Trained on Test Search Space), the MPHD models were pre-trained on the training super-dataset without the training dataset for the test search space, i.e., $\{(D_j^{\text{train}}, S_j)\}_{j=1}^N \setminus \{(D_i^{\text{train}}, S_i)\}$. Therefore, in the NToT setting, the pre-trained model of MPHD is tested on functions from an unseen search space. Because homogeneous meta BO methods need to be pre-trained in the same search space used for BO test, their models were pre-trained on $\{(D_i^{\text{train}}, S_i)\}$.

For PD1, we report the average normalized simple regret on all test sub-datasets in PD1, i.e., sub-datasets in $D_P^{\text{test}}$. Same as above, methods that are not pre-trained can be directly tested. As a special case of the NToT setting, when testing the BO performance of MPHD on PD1 Dataset, the MPHD model were pre-trained on the entire HPO-B Super-dataset $\{(D_j^{\text{train}}, S_j)\}_{j=1}^N$ but not on PD1 Dataset. In this case, MPHD needs to generalize to an unseen search space that is not even in the same super-dataset that it is trained on. Models of homogeneous meta BO methods were pre-trained on $D_P^{\text{train}}$, the training sub-datasets in PD1.

For Step ① in the pre-training of MPHD and the pre-training of meta-BO baseline methods HyperBO, FSBO, and ABLR, the GP parameters in every domain were fit by minimizing the NLL of training data in that domain (Eq. 3) using the Adam optimizer (Kingma & Ba, 2015; Wistuba & Grabocka, 2021). For Step ② in the pre-training of MPHD, the NN-based length-scale prior was optimized by minimizing Eq. 4 across all training domains using the Adam optimizer. Hyperparameters of Non-NN priors were directly fit using standard library functions for MLE of Gamma and Normal distributions. When using pre-trained MPHD models for BO, the GP was re-trained on the current observations at every BO iteration. The re-training process used the pre-trained model as a prior, aiming to approximate the posterior as defined in Eq. 6, and utilized the L-BFGS optimizer. It is noted that L-BFGS was recommended by Wang et al. (2022) as the standard objective optimization method for HyperBO, while Adam was recommended by Wistuba & Grabocka (2021) that applied the FSBO method on HPO-B. More details of the training processes are available in §E.2 and §F.

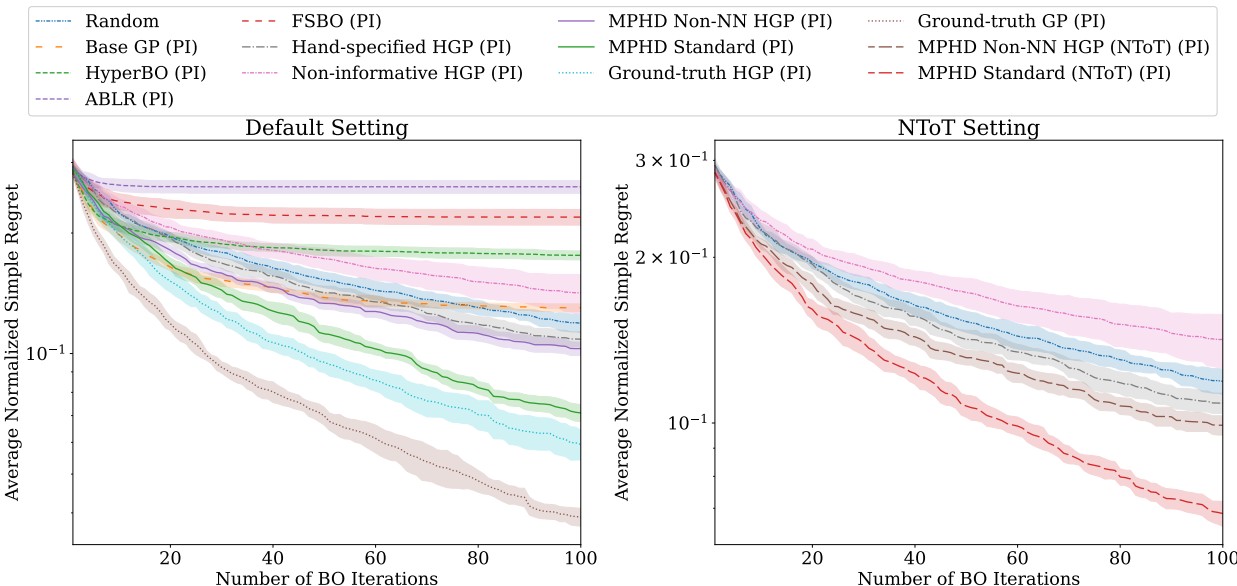

Figure 7: Results on Synthetic Super-dataset (L). The highlighted areas show mean $\pm$ std for each method. Left: Results on the default setting where all meta BO methods can use the training data in the test search space. MPHD Standard obtained a similar performance to Ground-truth HGP, showing that the model learned in MPHD is a good characterization of the ground truth prior. Note that in practice, BO methods do not have access to the ground-truth priors, so it is not realistic to expect that our method can outperform those two ground-truth methods. Right: the NToT setting and only methods that do not require pre-training on the test search space are included. MPHD Standard (NToT) outperformed other methods, demonstrating the ability of MPHD to generalize to unseen search spaces.

## 4.3 Results on Bayesian optimization

For all of the following experiments, the budget for BO is 100, and there are a set of 5 initial observations that are randomly sampled for each of the 5 random seeds. The acquisition function used for all GP-based methods is *Probability of Improvement* (PI) (Kushner, 1964) with target value $\max(y_t) + 0.1$. As shown by Wang & Jegelka (2017), PI can obtain high BO performance by setting good target values. Results on other acquisition functions are included in §F.

**Synethetic Super-dataset (T).** Fig. 7 (left) shows the BO performances of compared methods on Synthetic Super-dataset (L). MPHD Standard, which has an NN-based length-scale prior model, outperformed all the baselines except for the Ground-truth HGP and Ground-truth GP. This demonstrates the ability of MPHD to effectively learn a good prior model across heterogeneous search spaces during pre-training. Moreover, while Base GP should be more customized to the search space than MPHD Non-NN HGP, MPHD Non-NN HGP was able to achieve much better performance than Base GP; this shows the importance of Step ② in MPHD regardless of using domain-specific contexts. Another observation is that MPHD Standard outperformed MPHD Non-NN HGP by a large margin, demonstrating the effectiveness of using an NN to capture the dependence of domain-specific priors on the context features.

Interestingly, HyperBO achieved better performance in the initial few BO iterations but it fell behind other methods after about 10 BO iterations for the test functions in Synethetic Super-dataset (L). One possible reason is that the posterior GPs in HyperBO significantly deviated from the ground truth GP. As noted in Wang et al. (2022), the difference between the predicted posterior variance and the ground truth posterior variance increases with the number of observations. This can be a critical issue for complex datasets like Synthetic Super-dataset (L), though it was not a severe problem for HPO-B and PD1 (results below). FSBO and ABLR can be viewed as special cases of HperBO with zero mean function, which makes it difficult to obtain a good prior or posterior compared to the ground truth. On the contrary, the posterior inference in MPHD was done for a hierarchical GP instead of a single GP in HyperBO. Since a hierarchical GP can

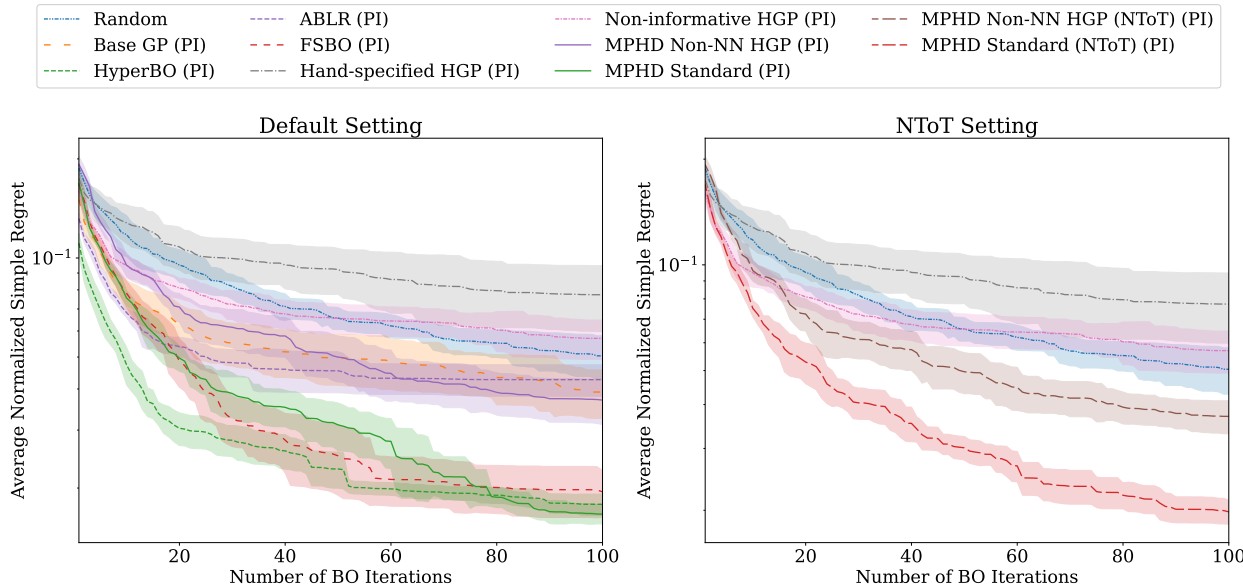

Figure 8: Results on HPO-B Super-dataset. Left: in the default setting, HyperBO had superior performance at first during the BO iterations, but MPHD Standard eventually achieved the lowest regret after 100 BO iterations. Right: in the NToT setting, MPHD Standard (NToT) achieved the best performance among all methods that do not require pre-training on the test search space.

typically assign probability density to a larger space of functions than a single GP, MPHD was able to achieve much more robust performance than HyperBO.

Fig. 7 (right) shows the BO results on Synthetic Super-dataset (L) of compared methods in the NToT setting where the search space used for BO test is not used for pre-training. HyperBO, ABLR, and FSBO cannot be tested in this setting as they require pre-training in homogeneous search spaces for each test function. MPHD Standard in the NToT setting achieved superior performance over baseline methods such as the Hand-specified HGP prior and the Non-informative HGP prior, which demonstrates the ability of MPHD to generalize its pre-trained model to functions in new search spaces that are unseen during pre-training. Similar to the results in Fig. 7 (left), MPHD Standard again outperformed MPHD Non-NN HGP, showing the importance of using domain-specific priors.

**HPO-B.** Fig. 8 (left) shows the BO performances on HPO-B Super-dataset. MPHD Standard achieved lower final regrets than all baseline methods. HyperBO had superior BO performance than MPHD Standard when the number of BO iterations was smaller than 80 but was eventually overtaken by MPHD Standard. As explained above, the performance plateau of HyperBO and FSBO could be related to the increase of the posterior approximation errors Wang et al. (2022). Similar to the observation on Synthetic Super-dataset (L), MPHD variant with an NN-based length-scale prior model outperformed MPHD Non-NN HGP.

Fig. 8 (right) shows the BO results on HPO-B Super-dataset of compared methods in the NToT setting. Notably, MPHD Standard (NToT) performed only slightly worse than MPHD Standard pre-trained in all search spaces even though MPHD Standard (NToT) was not trained in the search space it was tested in. The good performance of MPHD Standard (NToT) further demonstrates the capability of MPHD to generalize across search spaces in real-world problems. MPHD Standard (NToT) outperformed all other methods in this setting, again demonstrating its ability to generalize to new test functions in unseen search spaces.

**PD1.** Fig. 9 (left) shows the BO results of methods valid in the NToT setting on PD1 Dataset. Here the MPHD models were pre-trained on training datasets in HPO-B Super-dataset, but were not pre-trained on PD1. MPHD Standard (NToT) achieved the best performance in the NToT setting. Even though HPO-B and PD1 are two separately collected real-world hyperparameter tuning datasets, MPHD was capable of generalizing the model pre-trained on HPO-B to test functions in PD1. Fig. 9 (right) also includes the BO performances of the homogenous meta BO methods on PD1 Dataset. MPHD Standard (NToT) was

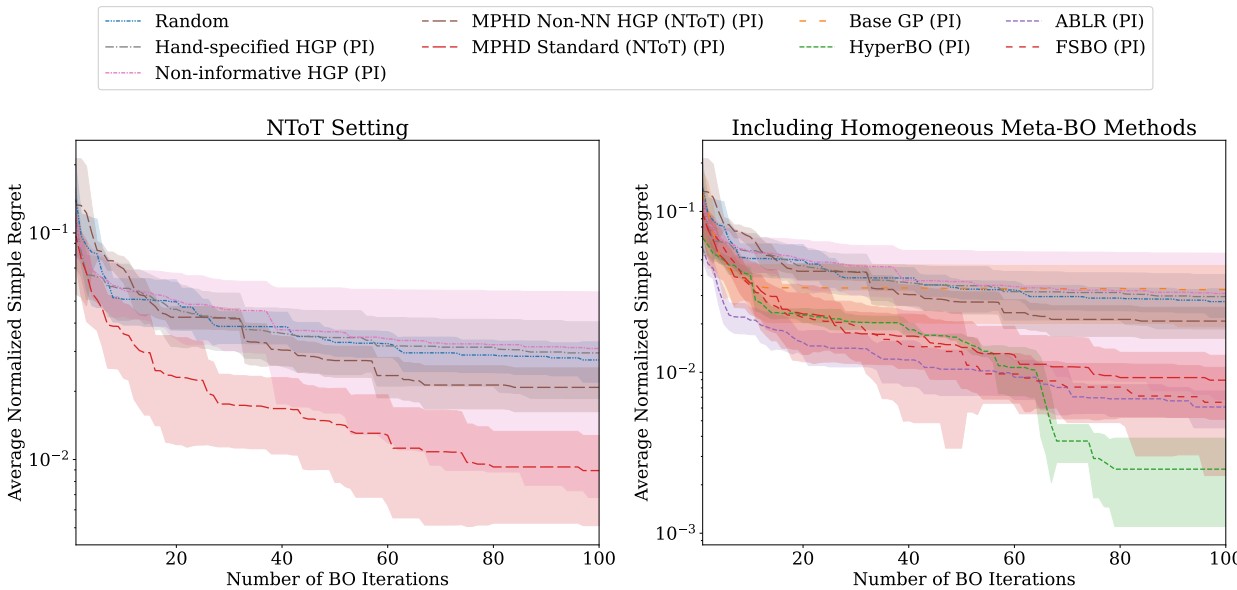

Figure 9: Results on PD1 Dataset. Left: in the NToT setting, MPHD Standard (NToT) achieved the best performance among all methods that do not require pre-training on the test search space, demonstrating its ability to generalize to unseen search spaces. Right: we include homogeneous meta BO methods in addition to NToT methods. Homogeneous meta BO methods HyperBO, ABLR, and FSBO outperformed MPHD Standard (NToT), which is expected since these methods were directly pre-trained on PD1 Dataset while MPHD Standard (NToT) was not.

outperformed by HyperBO, ABLR, and FSBO, which is not surprising as these single-search-space baselines were pre-trained on training sub-datasets in PD1 while MPHD Standard (NToT) was not.

## 5 Discussions and conclusions

In this work, we propose MPHD, the first GP-based transfer learning BO method that works on heterogeneous search spaces. The key idea is to pre-train a hierarchical GP with domain-specific priors on training data from functions in different domains. MPHD does not need access to BO trajectories in the format of an ordered list of data points. Instead, MPHD can effectively make use of an unordered set of datapoints as long as they are partitioned to different functions and different domains (i.e., a super-dataset in §2.1). Our theoretical and empirical analyses showed that MPHD enjoys appealing asymptotic properties and performs competitively on challenging hyperparameter tuning tasks, making it both a theoretically sound and a practical transfer learning method.

**Limitations and future work**

For a BO task, MPHD only learns the prior model in the form of a domain-specific hierarchical GP. While this allows a separation of model and decision making strategy, there are other components that can also be meta-learned, such as acquisition functions (Volpp et al., 2020), to maximize the effectiveness of the BO system. One direction of future work is jointly pre-train all components of BO to allow more flexibility and further improve the performance.

Like most machine learning methods, MPHD is subject to assumptions, including a stationary kernel function, a constant mean function and the availability of a super-dataset with domain-specific contexts. Possible future work includes relaxing assumptions on kernel and mean functions and incorporating architecture search to enrich the space of hierarchical GP priors. The assumptions on data are naturally satisfied in our experiments such as HPO-B, since our context encodings only require input dimensions and whether the input is continuous. However, for some other types of data, the domain-specific contexts might not be real

vectors. Our work builds a strong foundation for generalizing to those more complex contexts if they can be encoded as real vectors, and a future work is to work with those more complex domain-specific contexts.

**Broader impact**

Hyperparameter tuning for machine learning (ML) models, especially deep learning models, can be very costly if we repeatedly evaluate a large number of hyperparameters. Each single evaluation of a hyperparameter value requires training and testing a new instantiation of the model. Our framework MPHD, with its superior BO performance discussed in §4.3, can help to reduce the number of evaluations needed for hyperparameter tuning tasks and thus reduce their computational cost and carbon footprint potentially by a large margin.

**Acknowledgments**

We thank Richard Zhang for feedback, and Jasper Snoek and Eytan Bakshy for helpful comments on the previous version of this work (Fan et al., 2022), which was presented at NeurIPS 2022 Workshop on Gaussian Processes, Spatiotemporal Modeling, and Decision-making Systems. The key difference to this previous work is the inclusion of domain-specific hierarchical Gaussian processes as opposed to using a universal model. Our work also benefited from Microsoft Azure credits provided by the Harvard Data Science Initiative, as well as Google Cloud Platform Credit Awards provided by Google.

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

Table 1: Problem formulation notations

| Notation | Description |
|---|---|
| $\mathcal{X}^{(i)}$ | domain. |
| $F_i = \{f_{ij}\}_{j=1}^{M_i}$ | functions in domain $\mathcal{X}^{(i)}$. |
| $\mathcal{GP}_i = \mathcal{GP}(\mu_i, k_i; \theta_i)$ | Gaussian process that generates functions $F_i$ in domain $\mathcal{X}^{(i)}$. |
| $\mu_i$ | mean function of $\mathcal{GP}_i$. |
| $k_i$ | kernel function of $\mathcal{GP}_i$. |
| $\theta_i$ | parameters of $\mathcal{GP}_i$. |
| $\alpha_{ih}$ | hyperparameter (distribution parameters) for the $h$-th parameter $\theta_{ih}$ of $\mathcal{GP}_i$. |
| $S_i = [s_{ih}]_{h=1}^{H_i}$ | context for dataset $i$. |
| $\phi$ | model, i.e., mapping from $s_{ih}$ to $\alpha_{ih}$. |
| $D_{ij} = \{(x_{ij}^{(l)}, y_{ij}^{(l)})\}_{l=1}^{L_{ij}}$ | sub-dataset. |
| $D_i = \{D_{ij}\}_{j=1}^{M_i}$ | dataset. |
| $D = \{(D_i, S_i)\}_{i=1}^{N}$ | super-dataset. |
| $\bar{D} = \bar{D}_1 \cup \cdots \cup \bar{D}_{M_i}$ | augmented pseudo sub-dataset in Lemma 1. |
| $\bar{D} = \{(x^{(t)}, y^{(t)})\}_{t=1}^{T}$ | datapoints in the pesudo sub-dataset. |

## A    Table of notations

Table 1 provides a list of the notations used in this paper.

## B    Laplace approximation of integrals in the derivation of Eq.1

We derive Eq.1 as follows.

$$
\begin{aligned}
\mathcal{L}(\phi) &= \sum_{i=1}^{N} \log \int_{\theta_i} p(D_i \mid \theta_i) p(\theta_i \mid \phi, S_i) \, d\theta_i \\
&= \sum_{i=1}^{N} \log \int_{\theta_i} p(\theta_i \mid \phi, S_i) e^{\log p(D_i \mid \theta_i)} \, d\theta_i \\
&= \sum_{i=1}^{N} \log \int_{\theta_i} p(\theta_i \mid \phi, S_i) e^{\sum_{j=1}^{M_i} \log p(D_{ij} \mid \theta_i)} \, d\theta_i \\
&= \sum_{i=1}^{N} \log \int_{\theta_i} p(\theta_i \mid \phi, S_i) e^{M_i \left(-\frac{1}{M_i} \sum_{j=1}^{M_i} \log p(D_{ij} \mid \theta_i)\right)} \, d\theta_i.
\end{aligned}
$$

For every $i = 1, \ldots, N$, $i = 1, \ldots, N$, $\theta_i = [\theta_{ih}]_{h=1}^{H_i}$. For the following derivation, we assume that there is a reasonably large cap on every dimension of $\theta_i$ so that the domain of $\theta_i$ is a compact set $K_i \subset \mathbb{R}^{H_i}$. Let $h_i(\theta_i) = p(\theta_i \mid \phi, S_i)$ and $f_i(\theta_i) = -\frac{1}{M_i} \sum_{j=1}^{M_i} \log p(D_{ij} \mid \theta_i)$. Here $h_i$ is continuous on $K_i$ and $f_i$ is twice continuously differentiable on $K_i$. Let $\hat{\theta}_i = \arg\min_{\theta_i \in K_i} f_i(\theta_i) = \arg\max_{\theta_i \in K_i} p(D_i \mid \theta_i)$. $\hat{\theta}_i$ is a global minimizer of $f_i$ on $K_i$ and we have $h_i(\hat{\theta}_i) \neq 0$. We also assume that $f_i''(\hat{\theta}_i)$ is a positive definite matrix. When

$M_i \to \infty$ for $i = 1, \ldots, N$, we have

$$
\begin{aligned}
\mathcal{L}(\phi) &= \sum_{i=1}^{N} \log \int_{\theta_i} p(\theta_i \mid \phi, S_i) e^{M_i \left( -\frac{1}{M_i} \sum_{j=1}^{M_i} \log p(D_{ij}|\theta_i) \right)} \, \mathrm{d}\theta_i \\
&\approx \sum_{i=1}^{N} \log \int_{\theta_i \in K_i} h_i(\theta_i) e^{-M_i f_i(\theta_i)} \, \mathrm{d}\theta_i \\
&\approx \sum_{i=1}^{N} \log \sqrt{\frac{(2\pi)^{H_i}}{M_i^{H_i} \det f_i''(\hat{\theta}_i)}} h_i(\hat{\theta}_i) e^{-M_i f_i(\hat{\theta}_i)} \, \mathrm{d}\theta_i \\
&= \sum_{i=1}^{N} \log \sqrt{\frac{(2\pi)^{H_i}}{M_i^{H_i} \det \frac{\mathrm{d}^2}{\mathrm{d}\theta^2}(-\frac{1}{M_i} \sum_{j=1}^{M_i} \log p(D_{ij} \mid \theta_i))|_{\theta_i = \hat{\theta}_i}}} p(\hat{\theta}_i \mid \phi, S_i) p(D_i \mid \hat{\theta}_i) \\
&= \sum_{i=1}^{N} \log \sqrt{\frac{(2\pi)^{H_i}}{M_i^{H_i} \det \frac{\mathrm{d}^2}{\mathrm{d}\theta^2}(-\frac{1}{M_i} \log p(D_i \mid \theta_i))|_{\theta_i = \hat{\theta}_i}}} p(\hat{\theta}_i \mid \phi, S_i) p(D_i \mid \hat{\theta}_i).
\end{aligned}
\tag{7}
$$

Here the approximation step of Eq. 7 uses the Laplace approximation of integrals.

## C    More details of theoretical analysis

**Theorem 2 Discussion**    Recall from Lemma 1 that the the pseudo sub-dataset $\bar{D}$ satisfies the increasing-domain configuration. Its collection of input locations comes from the sub-datasets $\{D_{ij}\}_{j=1}^{M_i}$ (each $D_{ij}$ has $L_{ij}$ observations) and are denoted as $x^{(1)}, x^{(2)}, \cdots, x^{(T)}$ where $T = \sum_{j=1}^{M_i} L_{ij}$. Respectively, we denote the observations at those locations as $y^{(1)}, y^{(2)}, \cdots y^{(T)}$. It is obvious that as $M_i \to \infty$ we will have $T \to \infty$. Therefore, to show the asymptotic property of the maximum-likelihood estimator $\hat{\theta}_i$, we leverage the properties derived in previous work on the increasing domain setting. That is, we aim to show as $T \to \infty$, it is true that $\hat{\theta}_i \xrightarrow{p} \theta_i^*$.

Similar to Eq.(3), we define the observations in $\bar{D}$ as vector $\boldsymbol{y}^{(\theta_i)} = [(y^{(t)} - \mu_i(x^{(t)}; \theta_i)]_{t=1}^{T}$ and use $K^{(\theta_i)}$ to denote the invertible covariance matrix in $\bar{D}$: $K^{(\theta_i)} = [k_i(x^{(t)}, x^{(t')}; \theta_i)]_{t=1, t'=1}^{T}$. Let $\mathfrak{L}_T(\theta_i)$ be a function proportional to the *negative* log-likelihood of $T$ observations:

$$
\mathfrak{L}_T(\theta_i) = \frac{1}{T} \log(|K^{(\theta_i)}|) + \frac{1}{T} \left( \boldsymbol{y}^{(\theta_i)} \right)^\top \left( K^{(\theta_i)} \right)^{-1} \left( \boldsymbol{y}^{(\theta_i)} \right).
$$

Then the maximum likelihood estimator for $\theta_i$ is given by

$$
\hat{\theta}_i \in \arg\min_{\theta_i \in C_i} \mathfrak{L}_T(\theta_i).
$$

Suppose that assumptions (1)-(3) in section 3.1 hold for a zero-mean and anisotropic Matérn kernel whose length-scale parameter $\theta_i$ is to be estimated, we can derive the following lemma:

**Lemma 2.1** (proved as Lemma 2 in Bachoc (2021)): For any $\theta_i \in C_i$, as $T \to \infty$:

$$
\mathrm{var}(\mathfrak{L}_T(\theta_i)) = o(1).
$$

This lemma shows that for any $\epsilon > 0$, there exists $T$ large enough such that the variance of the likelihood function on the set of $T$ observations satisfies $\mathrm{var}(\mathfrak{L}_T(\theta_i)) < \epsilon$ in probability.

Then we proceed with the proof for Theorem 2 following Bachoc (2021)):

**Proof of Theorem 2 (Bachoc (2021) Theorem 1)**: From Lemma 2.1 we have that for any $\theta_i \in C_i$:

$$
\mathfrak{L}_T(\theta_i) - \mathbb{E}[\mathfrak{L}_T(\theta_i)] \xrightarrow{p} 0 \quad \text{as } T \to \infty.
$$

We can also obtain that

$$
\sup_{\theta_i \in C_i} |\mathfrak{L}_T(\theta_i) - \mathbb{E}[\mathfrak{L}_T(\theta_i)]| = o_p(1).
\tag{8}
$$

Previous work from Bachoc (2014) (Proposition 3.1) shows that there exists constant $A > 0$ such that for $\theta_i \in C_i$:

$$\mathbb{E}[\mathfrak{L}_T(\theta_i)] - \mathbb{E}[\mathfrak{L}_T(\theta_i^*)] \geq A\frac{1}{T}\sum_{t,t'=1}^{T} \left(k_i(x^{(t)}, x^{(t')}; \theta_i) - k_i(x^{(t)}, x^{(t')}; \theta_i^*)\right)^2. \tag{9}$$

Combining the above expression and the asymptotic identifiability condition, we obtain that for $\epsilon > 0$, there is a strictly positive constant $B$ such that for $T$ large enough,

$$\inf_{\theta_i \in C_i, ||\theta_i - \theta_i^*|| > \epsilon} (\mathbb{E}[\mathfrak{L}_T(\theta_i)] - \mathbb{E}[\mathfrak{L}_T(\theta_i^*)]) \geq B > 0. \tag{10}$$

With Eq.(8), (10) and since $\hat{\theta}_i$ belongs to the M-estimator class, according to Theorem 5.7 of Van der Vaart (2000) we conclude that as $M_i \to \infty$ we have $T \to \infty$ and thus the maximum likelihood estimator $\hat{\theta}_i \xrightarrow{p} \theta_i^*$. $\quad\square$

**Proof of Theorem 3** According to Theorem 2, as the number of sub-datasets $M_i \to \infty$, we have $\hat{\theta}_i \to \theta_i^*$, i.e. each maximum-likelihood estimator $\hat{\theta}_i$ is a consistent estimator of the ground truth GP parameter $\theta_i^*$. In the problem formulation we assume that each GP parameter $\theta_i$ is sampled *i.i.d.* from the prior distribution $\Theta(\alpha_{ih})$. Therefore, using previous results on properties of MLE given *i.i.d.* samples (Wackerly et al., 2014), we know that the maximum-likelihood estimator $\hat{\alpha}_{ih}$ in general satisfies the consistency and asymptotic normality properties. That is, with mild regularity constraint[3], as $N \to \infty$, we have $\hat{\alpha}_{ih} \xrightarrow{p} \alpha_{ih}^*$. $\quad\square$

Note that the MLE can fail to be consistent when certain regularity condition is violated. For instance, when the parameter is not identifiable, i.e. when multiple distinct $\alpha_{ih}^*$ exists.

Meanwhile, the authors of Karvonen & Oates (2022) pointed out that the MLE for the length-scale parameter is indeed ill-posed when the data is noiseless and can be unstable to small perturbations. However, they found that regularization with small additive Gaussian noise does guarantee well-posedness and this is equivalent to data corrupted by additive Gaussian noise. This is identical to our setting with the synthetic data and we found that in practice an accurate estimation for noise allows reliable inference for covariance parameters.

## D   Algorithm of MPHD

---

**Algorithm 1** MPHD pre-training and Bayesian optimization with acquisition function $ac(\cdot; \theta_f)$.

---

1: **function** MPHD $(D = \{(D_i, S_i)\}_{i=1}^{N}, f, S_f)$
2:      **for** $i = 1, \cdots, N$ **do**
3:          $\hat{\theta}_i \leftarrow$ PRE-TRAIN-STEP-①$(D_i)$
4:      **end for**
5:      $\hat{\phi} \leftarrow$ PRE-TRAIN-STEP-②$(\{\hat{\theta}_i, S_i\}_{i=1}^{N})$
6:      $D_f \leftarrow \emptyset$
7:      **for** $t = 1, \cdots, T$ **do**
8:          $\hat{\theta}_f = \arg\max_{\theta_f} p(\theta_f \mid D_f, S_f, \phi = \hat{\phi})$ (Eq. 6)
9:          $x_t \leftarrow \arg\max_{x \in \mathcal{X}^{(f)}} ac_t\left(x; \theta_f = \hat{\theta}_f\right)$
10:         $y_t \leftarrow$ OBSERVE$(f(x_t))$
11:         $D_f \leftarrow D_f \cup \{(x_t, y_t)\}$
12:      **end for**
13:      **return** $D_f$
14: **end function**

---

Algorithm 1 shows the model pre-training of MPHD and how the learned model $\hat{\phi}$ is used in BO on a new function $f$. Here the acquisition function $ac(\cdot; \theta_f)$ is a specific type of acquisition function (e.g., PI) associated

---

[3]For instance, this requires the log-likelihood function being twice differentiable w.r.t. $\alpha_{ih}$. More thorough discussions on other regularity conditions can be found in Lehmann & Casella (2006).

with a GP parameterized by $\theta_f$. The GP serves as the surrogate model for function $f$ during BO. While the ground-truth GP parameter $\theta_f$ for the function $f$ is unknown, the learned model $\hat{\phi}$ of MPHD is used to achieve an estimate $\hat{\theta}_f$ and the estimate is used to parameterize the GP associated with the acquisition function.

# E   More details of the experiment setups

Our code for the experiments is built upon the codebase of HyperBO (Wang et al., 2022) and is available at `https://github.com/Evensgn/hyperbo-mphd`.

## E.1   Details on the two synthetic super-datasets

Both synthetic super-datasets were generated using a constant mean function and an anisotropic Matérn kernel with a known smoothness parameter. In our setting, each dataset $D_i$ $(i = 1, \ldots, N)$ corresponds to $\mathcal{GP}_i$ parameterized by $\theta_i$, which includes the following parameters: constant mean (the value of the constant mean function), length-scale (which has the same number of dimensions as the domain $\mathcal{X}^{(i)}$), signal variance, and the noise variance.

For the prior distribution types of each of these GP parameters, we use a Normal distribution for constant mean and use a Gamma distribution for each of the remaining parameters. A Gamma distribution is parameterized by $a$ (shape) and $b$ (rate), while a Normal distribution is parameterized by $c$ (mean) and $d$ (standard deviation).

**Synthetic Super-dataset (S)** is a smaller super-dataset where the same GP prior is shared across all domains. It includes 20 datasets (domains) with 10 sub-datasets in each dataset. Each sub-dataset includes noisy observations at 300 input locations in its respective domain. The dimensions of the domains were randomly sampled between 2 and 5. As all the datasets are *i.i.d.* samples, we used the first 16 datasets to be training datasets and the remaining 4 datasets to be test datasets when testing the empirical asymptotic behavior of the pre-training of MPHD in §3.2.1. The underlying GP uses a constant mean function and an anisotropic Matérn kernel with smoothness parameter $\nu = 3/2$. The specific prior hyperparameters for the GP parameters are as follows: constant mean is sampled from Normal($c = 1.0, d = 1.0$), each dimension of length-scale is sampled from Gamma($a = 10.0, b = 30.0$), signal variance is sampled from Gamma($a = 1.0, b = 1.0$), and noise variance is sampled from Gamma($a = 10.0, b = 100000.0$).

**Synthetic Super-dataset (L)** is a larger super-dataset and has domain-specific GP priors. It includes 20 datasets (domains) with 20 sub-datasets in each dataset. Each sub-dataset includes noisy observations at 3000 input locations in its respective domain. The dimensions of the domains were randomly sampled between 2 and 14. For every dataset, we used 80% of its sub-datasets for training and the remaining 20% for testing. We used the first 16 datasets for training when testing the empirical asymptotic behavior of the pre-training of MPHD in §3.2.2, while all 20 datasets were used for experiments in §4. The underlying GP uses a constant mean function and an anisotropic Matérn kernel with smoothness parameter $\nu = 5/2$. The prior for GP parameter length-scale is domain-specific and the prior hyperparameters have a linear dependence on the number of domain dimensions. For a domain $\mathcal{X}^{(i)} \subset \mathbb{R}^{d_i}$, the length-scale GP parameter of every domain dimension is sampled from Gamma($a = 0.07692d_i + 0.8462, b = -0.3539d_i + 5.7077$). The priors for other GP parameters (constant mean, signal variance, and noise variance) are not domain-specific, and the specific hyperparameters as follows: constant mean is sampled from Normal($c = 0.5, d = 0.2$), signal variance is sampled from Gamma($a = 15.0, b = 100.0$), and noise variance is sampled from Gamma($a = 1.0, b = 10000.0$).

## E.2   Details on the compared methods

In this section, we provide additional details on the compared methods in §4.

MPHD Standard, MPHD Non-NN HGP, Base GP, Non-informative HGP, Hand-specified HGP, Ground-truth GP, and Ground-truth HGP all use a constant mean function and an anisotropic Matérn kernel with smoothness parameter $\nu = 5/2$.

HyperBO and FSBO use an anisotropic Matérn kernel ($\nu = 5/2$) with an MLP base, while ABLR uses a dot-product kernel with an MLP base. HyperBO uses a linear mean function with an MLP base. ABLR and FSBO use a zero mean function. The size of the MLP bases used for the mean and kernel in HyperBO, FSBO, and ABLR is (128, 128) on HPO-B and (32, 32) on Synthetic Super-dataset (L).

For the prior distribution types of each of these GP parameters, MPHD Standard, MPHD Non-NN HGP, Non-informative HGP, Hand-specified HGP all use a Normal distribution for constant mean and use a Gamma distribution for each of the remaining GP parameters. Ground-truth HGP also has the same distribution types as these are the distribution types used when generating Synthetic Super-dataset (L).

**MPHD Standard** uses an NN-based length-scale prior. The sizes of the hidden layers in the NN are (16, 16) and the activation function is hyperbolic tangent. The output layer is 2-dimensional, which represents the two hyperparameters of the prior distribution for length-scale (shape and rate of the Gamma distribution). The context $S_i = [s_{ih}]_{h=1}^{H_i}$ used in MPHD encodes information on whether every domain dimension is discrete or continuous and the total number of discrete and continuous dimensions in the domain $\mathcal{X}^{(i)}(i = 1, \ldots, N)$. For every domain dimension in a domain $\mathcal{X}^{(i)}$, the context $s_i h$ associated with the length-scale of this domain dimension is a 4-dimensional vector, where the first 2 dimensions are one-hot encoding to specify whether this domain dimension is discrete or continuous, and the remaining 2 dimensions of the context are the number of discrete and continuous dimensions in the domain $\mathcal{X}^{(i)}$, respectively. This context can be automatically constructed for any given dataset to be used by MPHD Standard.

**Hand-specified HGP** uses a hand-specified (and therefore could be misspecified) prior for each GP parameter, and the same prior is used for all domains. The specific hyperparameters are as follows: constant mean is sampled from Normal($c = 0.5, d = 0.5$), each dimension of length-scale is sampled from Gamma($a = 1.0, b = 0.1$), signal variance is sampled from Gamma($a = 1.0, b = 5.0$), and noise variance is sampled from Gamma($a = 1.0, b = 100.0$).

**Non-informative HGP** uses a Uniform distribution as the prior for each GP parameter. The specific hyperparameters are as follows: constant mean is sampled from Uniform($0.0, 1.0$), each dimension of length-scale is sampled from Uniform($0.00001, 30.0$), signal variance is sampled from Uniform($0.00001, 1.0$), and noise variance is sampled from Uniform($0.00001, 0.1$).

**Ground-truth GP** uses the ground-truth GP parameters of every domain in the Synthetic Super-dataset (L).

**Ground-truth HGP** uses the ground-truth prior hyperparameters for every GP parameter type in the Synthetic Super-dataset (L), including the domain-specific length-scale prior.

**Optimizations.** For Step ① in the pre-training of MPHD and the pre-training of meta-BO baseline methods HyperBO, FSBO, and ABLR, we fit the GP parameters in every domain by minimizing the NLL of training data in that domain using the Adam optimizer. The number of iterations for the Adam optimizer is 20000, and each sub-dataset is randomly sub-sampled to 50 observations at each iteration. The learning rate of Adam optimizer is 0.001. For Step ② in the pre-training of MPHD, the NN-based length-scale prior in MPHD Standard is also optimized using Adam optimizer. The number of iterations is 10000, and the learning rate is 0.001. Hyperparameters of Non-NN priors (including the priors for constant mean, signal variance, noise variance in MPHD Standard, and priors for all GP parameters in MPHD Non-NN HGP) are directly fit using standard library functions for MLE of Gamma and Normal distributions. When using pre-trained MPHD models for BO, the GP is re-trained on the current observations at every BO iteration. The re-training uses the pre-trained model as the prior and its purpose is to approximate the posterior defined in Eq. 6. The optimizer for this re-training is L-BFGS and the number of iterations is 100. Furthermore, L-BFGS was recommended by Wang et al. (2022) as the standard objective optimization method for HyperBO, while Adam was recommended by Wistuba & Grabocka (2021) that applied the FSBO method on HPO-B.

## E.3 BO Setups

When testing the BO performances of compared methods, the BO budget is 100, and 5 random observations are made prior to BO iterations for initialization. All methods are tested on 5 random seeds, while the set of

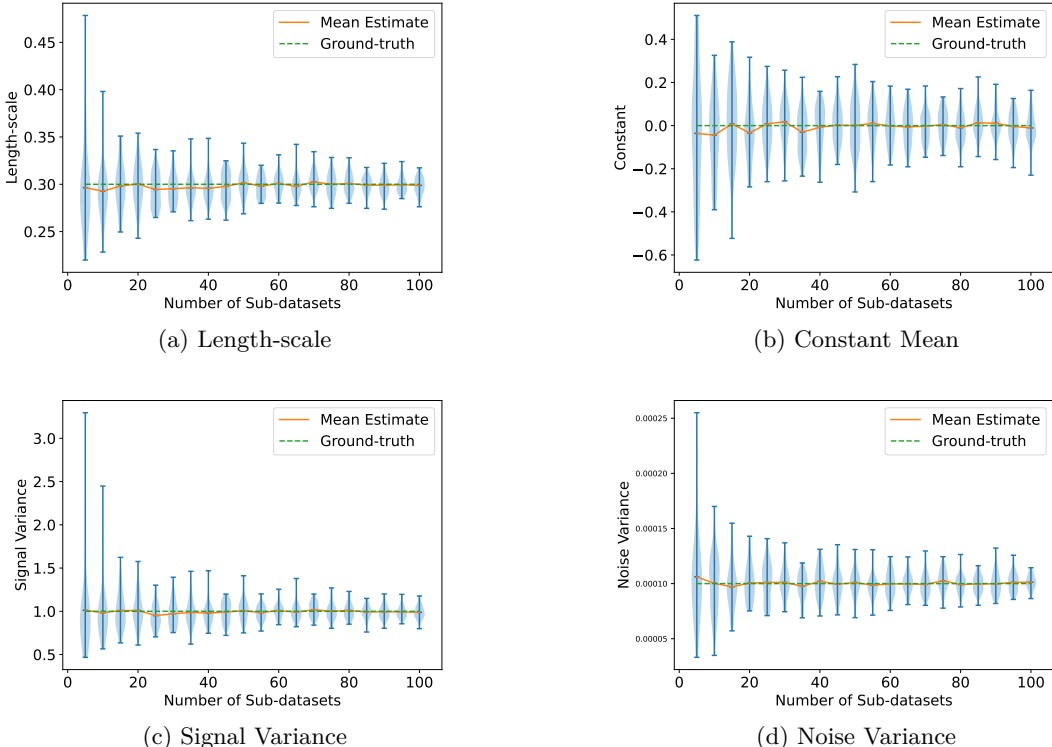

(a) Length-scale

(b) Constant Mean

(c) Signal Variance

(d) Noise Variance

Figure 10: Estimated length-scale, constant mean, signal variance, and noise variance parameters (distribution plotted over 50 random runs) of a 1-dimensional GP as the number of sub-datasets increases. Each sub-dataset has 25 observations. The variances of the estimated GP parameters decrease as the number of sub-datasets increases.

the initial 5 observations is the same for all methods given the same random seed. HPO-B Super-dataset provides 5 groups of initial observations, and each group contains 5 random observations for every test sub-dataset. Therefore, we just used the provided initial observations for BO on HPO-B. We randomly sampled 5 initial observations for each random seed and every test sub-dataset.

The PI acquisition function has parameter $\zeta = 0.1$, which means its target value is the maximum observation plus $\zeta$. The EI acquisition function uses the maximum observation as the target value. The GP-UCB acquisition function has parameter $\beta = 3$.

# F  Additional experiment results

In this section, we present additional empirical evidence on the asymptotic behaviors of the pre-training method of MPHD, as well as analyses on the performance of Bayesian optimization.

## F.1  Asymptotic behavior of fitting a single GP

Fig. 10 demonstrates the asymptotic behavior empirically for fitting the parameters of a single GP using an increasing number of sub-datasets with simulations on synthetic data generated with a 1-dimensional GP. The ground-truth GP parameters are shown in the figure. The variances of estimated GP parameters decrease as the number of sub-datasets increases.

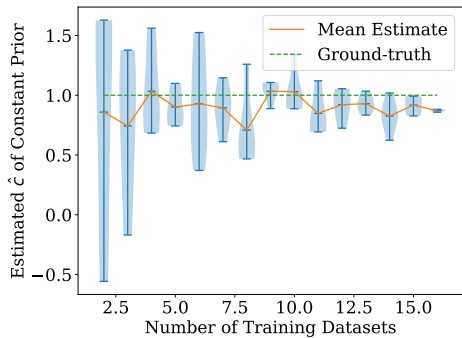

(a) Constant Mean - Mean parameter $c$ of Normal Distribution

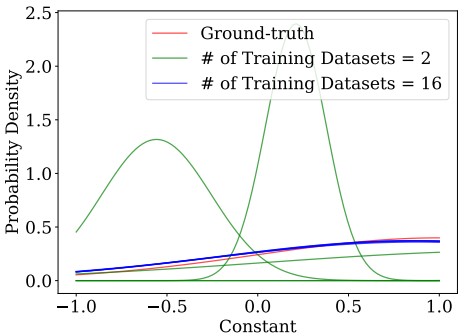

(b) Constant Mean - Normal Distribution

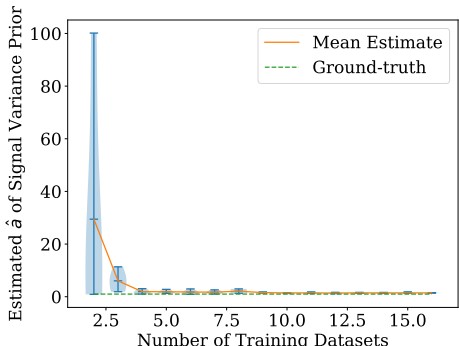

(c) Signal Variance - Shape parameter $a$ of Gamma Distribution

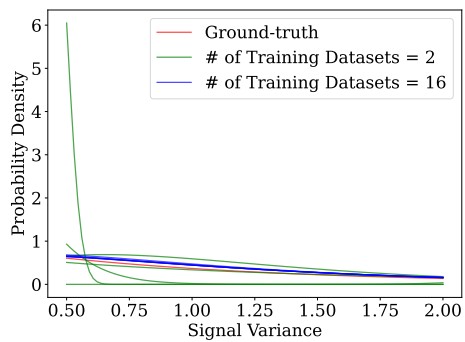

(d) Signal Variance - Gamma Distribution

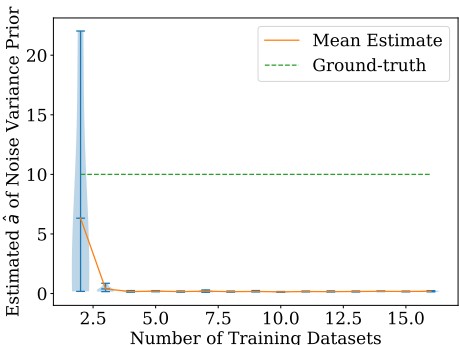

(e) Noise Variance - Shape parameter $a$ of Gamma Distribution

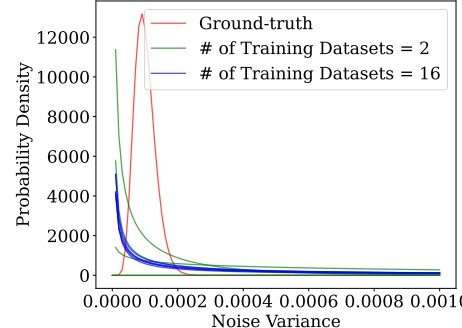

(f) Noise Variance - Gamma Distribution

Figure 11: For empirical asymptotic analysis on Synthetic Super-dataset (S) with a fixed one-dimensional length-scale prior, we plot on the left the estimated distribution parameter for GP parameters constant mean, signal variance, and noise variance as the number of training datasets increases, and on the right the probability density functions of the distributions that model each of the GP parameters for $N = 2$ and $N = 16$ together with the ground-truth distributions. Results with 5 random seeds are shown.

### F.2 Asymptotic behavior of the pre-training method of MPHD

Fig. 11 shows on the left the estimated prior distribution parameter (mean $c$ for Normal distribution and shape $a$ for Gamma distribution) for GP parameters including constant mean, signal variance and noise variance w.r.t. the number of training datasets (the results for length-scale are shown in Fig. 5 in the main paper). This is demonstrated only on Synthetic Super-dataset (S) since there is no known ground-truth prior for real-world super-datasets such as HPO-B.

In addition, the estimated prior distributions for each GP parameter with 2 training datasets and 16 training datasets along with the corresponding ground-truth prior distribution are shown on the right of Fig. 11 (the results for length-scale are shown in Fig. 5 in the main paper). For most of the GP parameters (length-scale, constant mean, and signal variance), the estimated prior distribution parameter shows decreasing variance and gets closer to the ground-truth value as the number of training datasets increases.

The estimated prior distribution also gets much closer to the ground-truth prior distribution when 16 training datasets are used compared to only using 2 training datasets. For the noise variance parameter, the $\alpha$ parameter of Gamma distribution is estimated to be a different value than the ground-truth value, and the estimated prior distribution shows a different shape than the ground-truth prior distribution even when using 16 training datasets. Note that each dataset of Synthetic Super-dataset (S) contains only 10 sub-datasets. This small number of sub-datasets in each search space is likely part of the reason for the difference between the estimated noise variance prior and the ground-truth prior. The variance of the estimated $\alpha$ parameter of Gamma distribution for noise variance still decreases as the number of training datasets increases. The visualization shows that the estimated prior distributions for noise variance also put most of the probability density over values between $[0, 0.0002]$ as the ground-truth prior distribution does despite the difference in their shapes.

### F.3 BO performances with different acquisition functions

§4.3 shows the BO results with the *Probability of Improvement* (PI) acquisition function, and here we present additional BO results with other acquisition functions including *Expected Improvement* (EI) and *GP-UCB* (UCB).

**Synethetic Super-dataset (L).** Fig. 12 (left) shows the BO performances of compared methods on Synthetic Super-dataset (L) with acquisition functions UCB and EI. Echoing the results with PI in §4.3, MPHD Standard outperformed all the baselines except for the Ground-truth HGP and Ground-truth GP with either UCB or EI as the acquisition function. Moreover, the final regret of MPHD Standard matches that of the Ground-truth GP, which is rather impressive.

Fig. 12 (right) shows the BO results on Synthetic Super-dataset (L) of compared methods with UCB and EI in the NToT setting where the search space used for BO test is not used for pre-training. With either UCB or EI, MPHD Standard (NToT) outperformed baseline methods in the NToT setting. It can also be observed that the performance MPHD Standard dominates MPHD Non-NN HGP in both the default setting and the NToT setting, with either UCB or EI.

**HPO-B.** Fig. 13 (left) shows the BO performances on HPO-B Super-dataset with acquisition functions UCB and EI. With EI as the acquisition function, MPHD Standard achieved lower final regrets than all baseline methods. With UCB as the acquisition function, HyperBO has a better performance than MPHD Standard. Notice that MPHD aims to recover the ground-truth prior distribution, and as we have shown here, we need a good acquisition function to unleash the best performance of BO given a pre-trained prior. Using misspecified priors, some sub-optimal acquisition functions may actually lead to better performance than an acquisition function that achieves better results with the ground-truth prior.

Fig. 13 (right) shows the BO results on HPO-B Super-dataset of compared methods with UCB and EI in the NToT setting. With either UCB or EI as the acquisition function, MPHD Standard (NToT) outperformed all other methods in the NToT setting. Similar to the observation on Synthetic Super-dataset (L), MPHD variant with an NN-based length-scale prior model outperformed MPHD Non-NN HGP.

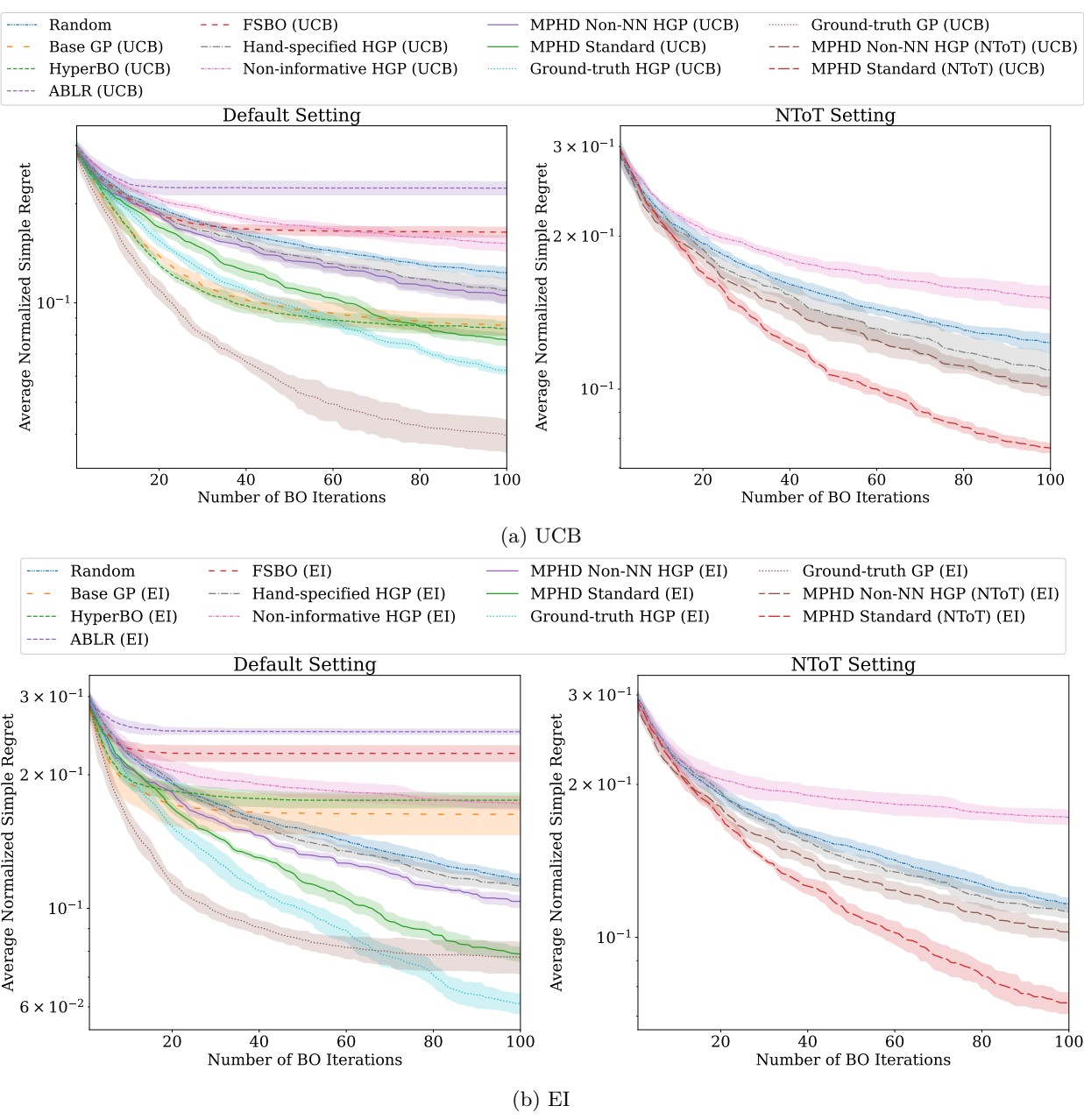

Figure 12: Average normalized simple regret of each method during BO for the Synthetic Super-dataset (L) in the two settings. The averages were taken over 5 random seeds, and the highlighted areas show mean $\pm$ std for each method. Results with the UCB and EI acquisition functions for each method are shown in respective rows. The figure on the left of each row shows the results in the default setting. The figure on the right of each row shows the results in the NToT setting and only methods that do not require pre-training on the test search space are included.

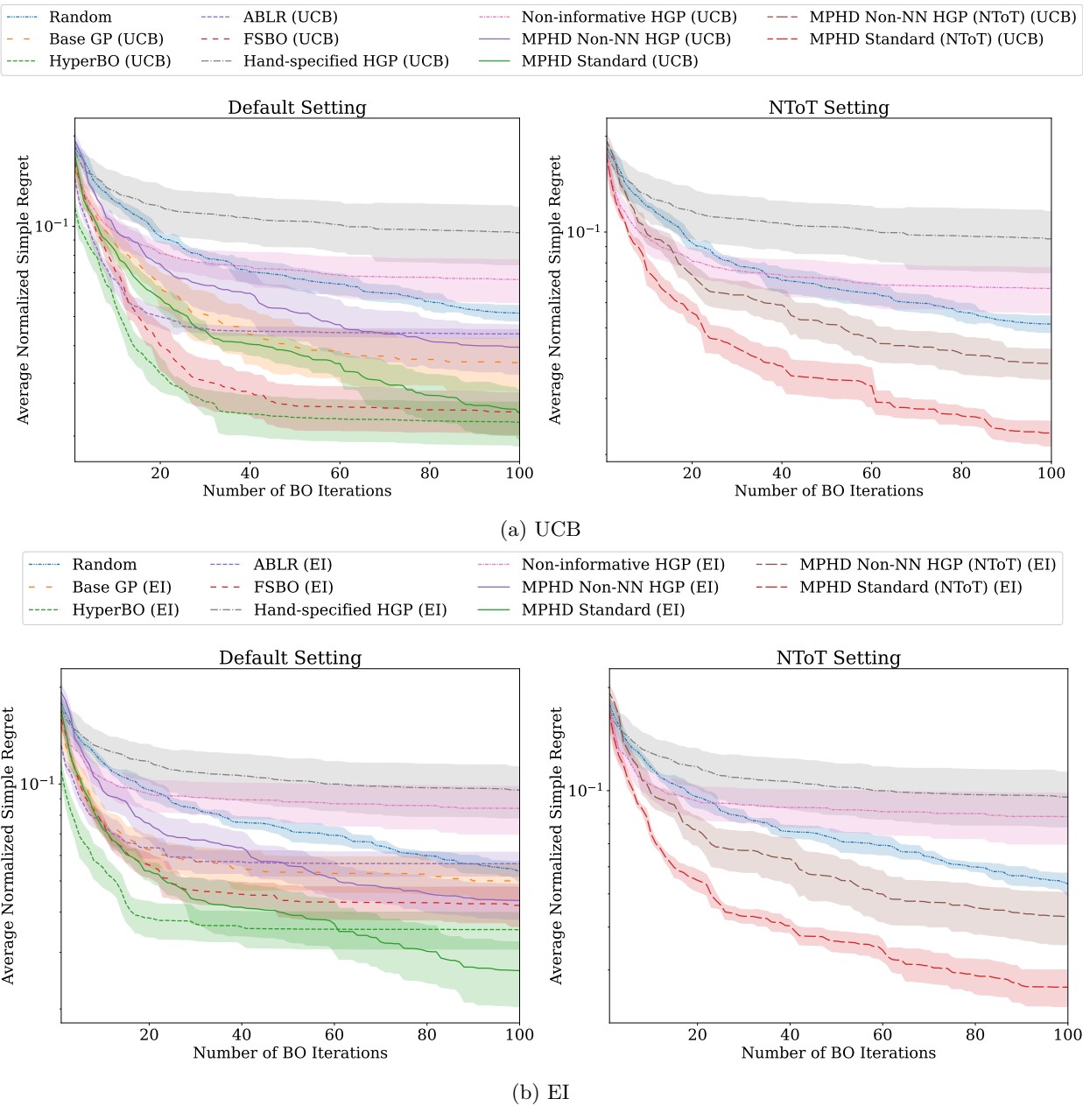

Figure 13: Average normalized simple regret of each method during BO for the HPO-B Super-dataset in the two settings. The averages were taken over 5 random seeds, and the highlighted areas show mean ± std for each method. Results with the UCB and EI acquisition functions for each method are shown here. The figure on the left of each row shows the results in the default setting. The figure on the right of each row shows the results in the NToT setting and only methods that do not require pre-training on the test search space are included.

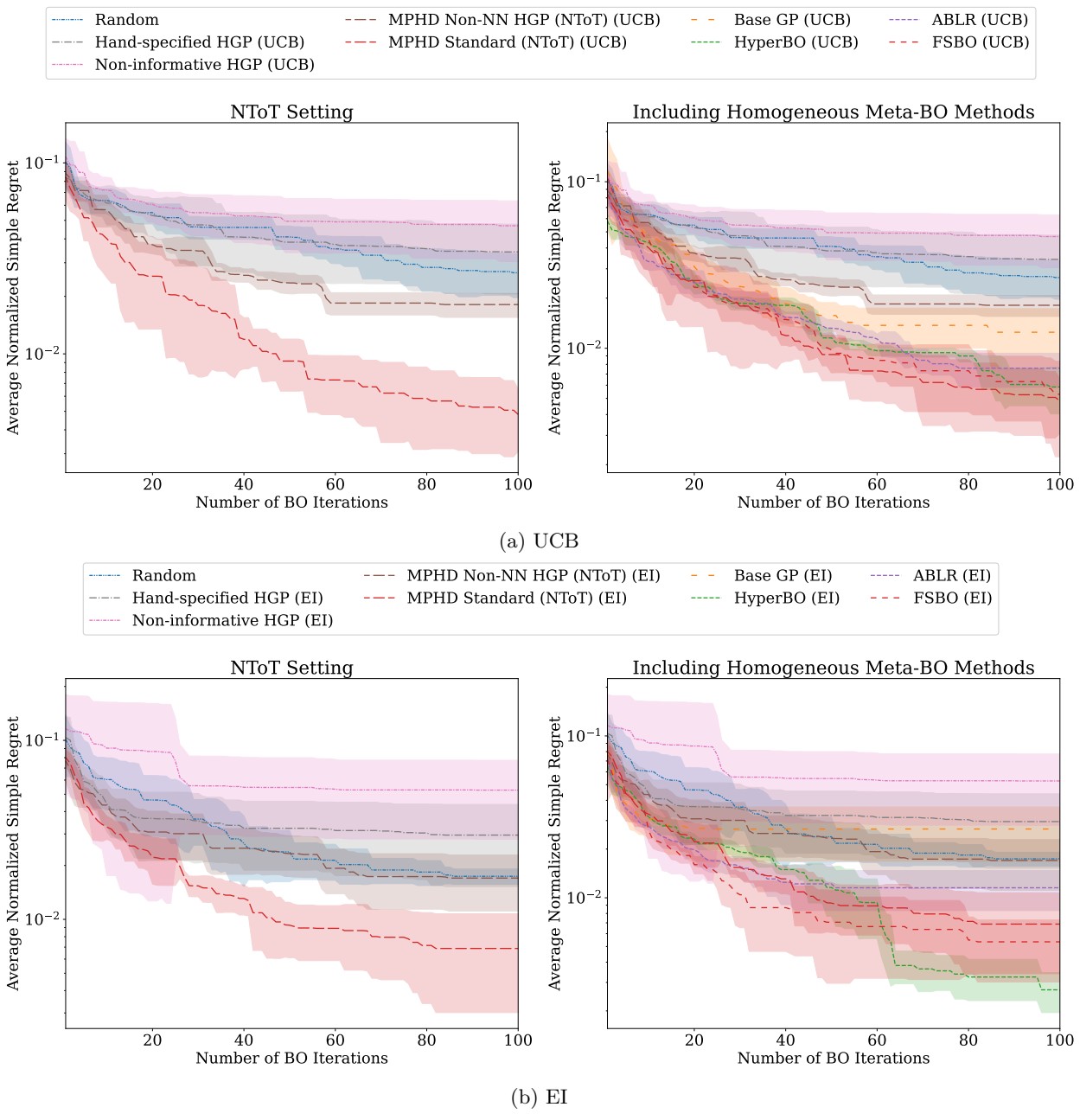

Figure 14: Average normalized simple regret of each method during BO for the PD1 Dataset. The averages were taken over 5 random seeds, and the highlighted areas show mean ± std for each method. Results with the UCB and EI acquisition functions for each method are shown here. The figure on the left of each row shows the results in the NToT setting. The figure on the right of each row includes homogeneous meta-BO methods in addition to methods that do not require pre-training on the test search space.

**PD1.** Fig. 14 (left) shows the BO results of methods valid in the NToT setting on PD1 Dataset with acquisition functions UCB and EI. Here the MPHD models were pre-trained on training datasets in HPO-B Super-dataset, but were not pre-trained on PD1. With either UCB or EI as the acquisition function, MPHD Standard (NToT) achieved the best performance in the NToT setting. Fig. 14 (right) also includes the BO performances of the homogenous meta BO methods on PD1 Dataset. With UCB as the acquisition function, MPHD Standard (NToT) outperformed all methods including single-search-space baselines HyperBO, ABLR, and FSBO. With the EI acquisition function, HyperBO and FSBO outperformed MPHD Standard (NToT), which is not surprising as HyperBO and FSBO were pre-trained on training sub-datasets in PD1 while MPHD Standard (NToT) was not.

In sum, we can see that with any of the three acquisition functions PI, UCB, and EI, MPHD generally outperforms the baselines (except the Ground-truth GP and Ground-truth HGP) in most cases across the two super-datasets and PD1, and also across the default setting and the NToT setting. This further demonstrates the effectiveness of the pre-training of MPHD and its ability to generalize to new test functions in both seen and unseen search spaces.

