# OpenReview forum: "Transfer Learning for Bayesian Optimization on Heterogeneous Search Spaces"
_TMLR — Accepted by TMLR_

### Review · Reviewer_YACK · 2023-11-09

**Summary Of Contributions:**

The paper proposes a method to perform transfer learning for Bayesian optimization when the search spaces of different BO tasks are potentially heterogeneous. The proposed method uses a neural network to learn the mapping from the contexts of different domains to the corresponding hyperparameters for the prior distribution from which the GP hyperparameters for this domain are sampled from. The proposed method is theoretically shown to be asymptotically consistent (i.e., the method is guaranteed to be able to find the true hyperparameters as the available datasets increase), and is empirically shown to perform competitively in experiments with both homogeneous and heterogeneous domains.

**Audience:**

Yes

**Broader Impact Concerns:**

No ethical concerns that I can think of.

**Claims And Evidence:**

Yes

**Requested Changes:**

In addition to the few weaknesses above, I have a couple of questions below:
- About the assumption of Asymptotic Identifiability, I wonder what's the intuition behind this assumption and how realistic is it? What's the requirement on the input locations in the sub-datasets in order to satisfy this assumption?
- The neural network $\phi$ outputs the hyperparameters $\alpha_{i,h}$ of the prior distribution, but different domains may have different number of such hyperparameters $\alpha_{i,h}$, so I wonder how do you decide the number of outputs from this neural network $\phi$?
- Figure 9, right figure, it looks like you didn't compare with MPHD Standard in this experiment?

**Strengths And Weaknesses:**

Strengths:
- The proposed method makes intuitive sense and is a natural method to handle the problem of heterogeneous domains in transfer learning for Bayesian optimization.
- Section 2 and Figure 1 gave nice and intuitive introduction to the proposed problem formulation and algorithm.
- I find the connection between the analyzed setting with increasing number of sub-datasets and the increasing domain setting (formalized by Lemma 1) very interesting.
- The experiments in Section 3.2 provide nice illustrations with regard to the behavior of the proposed method, which match well with expectations.
- The experimental results look good, the proposed methods not only work well in the novel setting of heterogeneous domains but are also competitive in the conventional problem setting with homogeneous domains.

Weaknesses:
- I think the paper lacks explanation on how to design the contexts for different domains. The contexts are important for the proposed method because if I understand correctly, they contain information about how the heterogeneous domains are related to each other, and hence they can be seen as the reason why information transfer across domains is possible. So, the design of the contexts is important for the paper and hence should be elaborated on.
- First paragraph of Section 3.1, it says here that the contexts for different domains are assumed to be the same in the theoretical analysis, does it mean that the theoretical analysis does not account for domain heterogeneity? If this is the case, then the significance of the theoretical contributions may be restricted because being able to handle heterogeneous domains is the main focus and novelty of this work.

---

> ### Author Response · Authors · 2023-11-27
>
> Thank you for your insightful and positive feedback. We are glad that you found our method intuitive, natural, and competitive in experiments with both homogeneous and heterogeneous domains. We are encouraged that you found our introduction to the problem formulation and method intuitive and that you found the connection to the increasing domain setting in Lemma 1 interesting. We address your specific comments below and have incorporated your feedback in the revised paper.
>
> > 1. I think the paper lacks explanation on how to design the contexts for different domains. The contexts are important for the proposed method because if I understand correctly, they contain information about how the heterogeneous domains are related to each other, and hence they can be seen as the reason why information transfer across domains is possible. So, the design of the contexts is important for the paper and hence should be elaborated on.
>
> Please see the general response on domain-specific contexts.
>
> > 2. First paragraph of Section 3.1, it says here that the contexts for different domains are assumed to be the same in the theoretical analysis, does it mean that the theoretical analysis does not account for domain heterogeneity? If this is the case, then the significance of the theoretical contributions may be restricted because being able to handle heterogeneous domains is the main focus and novelty of this work.
>
> The assumption in Section 3.1 that GP lengthscales of all domain dimensions share the same context does mean that our theoretical analysis is under a simplified version of the most general setting, however, we would like to point out that **the theoretical analysis (Theorem 3) still accounts for domain heterogeneity even under this simplification** as the different domains can still have different number of dimensions and different domain dimensions (either in the same domain or across different domains) can still have different GP lengthscales. For instance, if there is a 3-dimensional domain A and a 5-dimensional domain B, this assumption means that for the MPHD model in the theoretical analysis the GP lengthscales for the 3 dimensions of domain A and the 5 dimensions of domain B are assumed to be samples from the same one-dimensional prior distribution, but the 3+5=8 per-dimension GP lengthscales can all have different values. In the most general setting without this assumption, such as in our experiments in Section 4, the 8 per-dimension GP lengthscales can have different contexts and therefore each have their own prior distribution. Here in Section 3.1 we use this assumption to help simplify the theoretical analysis while still capturing domain heterogeneity.
>
> > 3. About the assumption of Asymptotic Identifiability, I wonder what's the intuition behind this assumption and how realistic is it? What's the requirement on the input locations in the sub-datasets in order to satisfy this assumption?
>
> Identifiability condition is required to show any consistency results of estimators. In our case it means that no two distinct covariance parameters exist such that the two covariance functions are the same on the set of randomly sampled points. The asymptotic identifiability is used in the case of increasing domain setting, where the sample size increases and the observations do not lie on a compact region. There is not any specific requirement on input locations for this assumption to hold. In our case, as the number of observations $T \rightarrow \infty$ in an unbounded region (increasing domain), it is realistic to expect that no two covariance parameters will lead to the exact same covariance functions. We have added those intuitions to the main paper (right after where the assumption is introduced).
>
> **(1/2)**

---

> ### Author Response · Authors · 2023-11-27
>
> > 4. The neural network $\phi$ outputs the hyperparameters $\alpha_{i,h}$ of the prior distribution, but different domains may have different number of such hyperparameters , so I wonder how do you decide the number of outputs from this neural network?
>
> The neural network $\hat{\phi}$ of MPHD outputs the hyperperameters $\alpha_{i,h}$ of the prior distribution for **a single GP parameter** (the $h$-th GP parameter of domain $i$), for example, the length-scale parameter for a specific domain dimension. For the same class of prior distribution, $\alpha_{i,h}$ for any domain dimension is of fixed length. For example, if the prior distribution for the length-scale GP parameter is chosen to be a gamma distribution, $\alpha_{i,h}$ for the length-scale of any domain dimension would be two-dimensional, representing the shape and scale parameters of that gamma distribution. Therefore, the neural network has a fixed output size. If a domain has $5$ dimensions, the length-scale GP parameter would also be of 5 dimensions, and in this case we would perform inference of the neural network $\hat{\phi}$ 5 times to generate the per-dimension prior hyperparameters $\alpha_{i,h}$ for the length-scale GP parameter of each domain dimension. This process is also illustrated in Fig. 1 (originally Fig. 6), where on the right side the pre-trained neural network is used in a test domain T.
>
> > 5. Figure 9, right figure, it looks like you didn't compare with MPHD Standard in this experiment?
>
> Figure 9 shows the experiment results on the PD1 Dataset. In our experiments we use the Synthetic Super-dataset and HPO-B Super-dataset as the two super-datasets to do comprehensive comparisons between all versions of MPHD and baselines. On the side, we use PD1 Dataset as an extra test dataset for the MPHD model pre-trained on HPO-B to demonstrate the capability of MPHD to generalize its model pretrained on one real-world super-dataset (HPO-B) to a different real-world dataset (PD1) that is unseen during pre-training. With this specific emphasize on transferring across different real-world datasets (and therefore also domains) as the purpose of the PD1 experiments, we include the MPHD version that is only pre-trained on HPO-B but not on PD1 at all [MPHD Standard (NToT)] to compare with baselines. We did not include MPHD Standard (which would be a MPHD model pre-trained on both HPO-B and PD1) in the PD1 experiments because it does not fit in the setting of generalizing to an unseen domain that we want to showcase with PD1, and comparisons of MPHD Standard and MPHD Standard (NToT) are already included in experiments on the two super-datasets. For the similar reason, MPHD Non-NN (NToT) is included in the PD1 experiments while MPHD Non-NN HGP is not.
>
> **(2/2)**

---

### Review · Reviewer_MKMS · 2023-11-11

**Summary Of Contributions:**

The paper develops a new transfer learning method, namely MPHD (model pre-training on heterogeneous domains) for Bayesian optimization on heterogeneous search space. The main idea is to pre-train a model that map from domain-specific context to the specifications of hierarchical GPs. The paper derives theoretical analysis to show that under some assumptions, the GP estimates (length scales) converge to the true value. Experimental results are conducted on various synthetic and real-world datasets to evaluate the effectiveness of the proposed method.

**Audience:**

Yes

**Claims And Evidence:**

No

**Requested Changes:**

+ [Major] The concept of “context” needs to be formulated rigorously and described in detail.
+ [Major] The assumptions used in all the lemmas and theorems in the paper need to be elaborated more to see if they’re common assumptions and whether they could occur in practice. All the assumptions need to be listed in the main paper.
+ [Major] The proofs of Theorems 1 and 2 need to be rigorously proven.
+ [Minor] The colours of the plots could be modified so as to help readers to see the performance of the proposed method and the baselines easier.

**Strengths And Weaknesses:**

# Strengths:
+ The approach of using previous datasets that include the optimization data of related optimization problems to aid the optimization of a new problem is important and is worth to investigate.
+ The paper’s idea of using these prior datasets to pre-train a probabilistic model so that for any new domain, the model can generate the prior distributions over the GP parameters to construct a domain-specific hierarchical GP, is interesting and seems reasonable.

+ The paper derives theoretical analysis to show that under some assumptions the proposed method can find the true GP hyperparameters when the number of sub-datasets converges to infinity (although I have various questions regarding the theoretical analysis).

+ The paper conducts experiments on both synthetic and real-world datasets and shows that the performance of the proposed method is better than existing baselines


# Weaknesses:
+ The concept of “context” (S) is very important and is used throughout the paper but it is never explained or defined in detail. There are examples of the context in Figure 2 but it’s hard to catch the definition of this concept from these examples. I’m really confused what exactly “context” means here.

+ The paper uses a lot of assumptions in order to derive the theorems presented in the paper, however, the implications of these assumptions are not described in detail. Are these common assumptions used in related research? Do these assumptions occur in practice? Some assumptions are really unclear me, for example, in the first paragraph in Section 3.1., it assumes that the contexts s_{ih} are the same across domain X^{(i)}, but there is definition of context, so I don’t really understand what this assumption means in practice. Furthermore, there are various assumptions used in the proof (in the appendix – Section C) but these are not mentioned in the paper – they should be explicitly mentioned in the paper.

+ The proofs of Theorems 2 & 3 are very vague. There are only 2-3 lines for the proof of each theorem. Even when we use some theoretical results from other papers, we still need to write again those results, and explain clearly with rigorous mathematical formulas on how we can apply those results here.

+ In the experiments, in the first sentence of Page 9, it says that for the two real-world datasets,
every domain dimension is normalized between, will this make all the search spaces of all the datasets to be the same? If yes, then these datasets might not be appropriate to be used to evaluate the proposed method as the method claims to work for different search spaces?
+ The colors of the plots make it quite difficult to distinguish between the methods. Besides, in Figure 7 (left plot – default setting), it seems the proposed method does not perform well?

---

> ### Author Response · Authors · 2023-11-27
>
> Thank you for your thoughtful feedback. We are glad that you found our problem setting important and worthwhile to investigate. We are encouraged that you found our method interesting, reasonable, and analysed both theoretically and empirically. We address your specific comments below and have incorporated your feedback in the revised paper.
>
> > 1. The concept of “context” (S) is very important and is used throughout the paper but it is never explained or defined in detail. There are examples of the context in Figure 2 but it’s hard to catch the definition of this concept from these examples. I’m really confused what exactly “context” means here. Related requested change: [Major] The concept of “context” needs to be formulated rigorously and described in detail.
>
> Please see the general response on domain-specific contexts.
>
> > 1.1 The paper uses a lot of assumptions in order to derive the theorems presented in the paper, however, the implications of these assumptions are not described in detail. Are these common assumptions used in related research? Do these assumptions occur in practice? Some assumptions are really unclear me, for example, in the first paragraph in Section 3.1, it assumes that the contexts $s_{ih}$ are the same across domain $X^{(i)}$, but there is definition of context, so I don’t really understand what this assumption means in practice.
>
> Firstly, we would like to address your specific question about the assumption in the first paragraph in Section 3.1. The assumption in Section 3.1 that GP lengthscales of all domain dimensions share the same context means that our theoretical analysis is under a simplified version of the most general setting but it still accounts for domain heterogeneity even under this simplification as the different domains can have different number of dimensions and different domain dimensions (either in the same domain or across different domains) can have different GP lengthscales. For instance, if there is a 3-dimensional domain A and a 5-dimensional domain B, this assumption means that for the MPHD model in the theoretical analysis the GP lengthscales for the 3 dimensions of domain A and the 5 dimensions of domain B are assumed to be samples from the same one-dimensional prior distribution, but the 3+5=8 per-dimension GP lengthscales can all have different values. In the most general setting without this assumption, such as in our experiments in Section 4, the 8 per-dimension GP lengthscales can have different contexts and therefore each have their own prior distribution. Here in Section 3.1 we use this assumption to help simplify the theoretical analysis while still capturing domain heterogeneity.
>
> > 2.2 Furthermore, there are various assumptions used in the proof (in the appendix – Section C) but these are not mentioned in the paper – they should be explicitly mentioned in the paper. Related requested change: [Major] The assumptions used in all the lemmas and theorems in the paper need to be elaborated more to see if they’re common assumptions and whether they could occur in practice. All the assumptions need to be listed in the main paper.
>
> All the assumption used in the proof and appendix has been listed in the main paper theory section. For assumptions (1)-(3) in Section 3.1, assumption (1) is the minimum spacing constraint and is required for showing the asymptotic consistency of MLE in the increasing domain setting [1]. Since our setup is equivalent to the increasing domain setting as shown in Lemma 1 this condition is satisfied. Assumption (2) considers the well-posedness of the estimation problem and assumption (3) requires that the model parameter can be uniquely determined. Those two assumptions are commonly required in analyzing the asymptotic behavior of maximum-likelihood estimators ([2], [3]).
>
> We have added some more intuitions related to those assumptions and pointers to related work in more general settings in the appendix.
>
> Reference:
>
> * [1] Bachoc, François. Asymptotic analysis of the role of spatial sampling for covariance parameter estimation of Gaussian processes. Journal of Multivariate Analysis 125 (2014): 1-35.
>
> * [2] Redner, Richard A., and Homer F. Walker. Mixture densities, maximum likelihood and the EM algorithm. SIAM review 26.2 (1984): 195-239.
>
> * [3] Wasserman, Larry. All of statistics: a concise course in statistical inference. Vol. 26. New York: Springer, 2004.
>
> **(1/2)**

---

> ### Author Response · Authors · 2023-11-27
>
> > 3. The proofs of Theorems 2 and 3 are very vague. There are only 2-3 lines for the proof of each theorem. Even when we use some theoretical results from other papers, we still need to write again those results, and explain clearly with rigorous mathematical formulas on how we can apply those results here. Related requested change: [Major] The proofs of Theorems 1 and 2 need to be rigorously proven.
>
> We agree with your request for more detailed and rigorous proofs of Theorems 2 and 3 (It seems there is a typo in your requested change with the theorem numbers, but we understand that you meant Theorems 2 and 3, not Theorems 1 and 2). We have included more detailed proofs for Theorems 2 and 3 in Appendix C of the revised paper.
>
> > 4. In the experiments, in the first sentence of Page 9, it says that for the two real-world datasets, every domain dimension is normalized between, will this make all the search spaces of all the datasets to be the same? If yes, then these datasets might not be appropriate to be used to evaluate the proposed method as the method claims to work for different search spaces?
>
> We would like to kindly point out that normalization of the domain dimensions does not change the structure of the BO problem and it does not negatively impact the generality of the tested BO algorithms including the proposed MPHD method. For any new test domain, the range of every domain dimension is known and we can always apply this normalization before running the BO algorithms, and the good x value found in the normalized domain can be simply mapped back to the original domain. This normalization is also the standard setting used in the HPO-B Super-dataset (Pineda-Arango et al., 2021).
>
> The domain heterogeneity in our problem setting comes from the non-trivial structural differences of different domain dimensions instead of the difference in ranges of different domain dimensions, as the range differences can be trivially solved by normalizing all domain dimensions. The more important non-trivial structural differences remain after this normalization of domain dimensions, and those structural differences are the challenges we aim to solve in the paper with the proposed MPHD method.
>
> > 5. The colors of the plots make it quite difficult to distinguish between the methods. Related requested change: [Minor] The colours of the plots could be modified so as to help readers to see the performance of the proposed method and the baselines easier.
>
> Thanks for suggesting updated the colors of the plots to help readers see the performances of different methods better. We have adjusted the colors and introduced additional line styles in Figures 7, 8, 9, 12, 13, 14 in the revised paper and we hope the modifications would help the readers read the results more clearly.
>
> > 6. In Figure 7 (left plot – default setting), it seems the proposed method does not perform well?
>
> It seems there is an misunderstanding of the results here. **The left plot of Fig. 7 actually shows that the proposed MPHD has the best performance across all the compared methods that do not have access to the ground-truth prior.** In the left plot in Fig. 7, the proposed method MPHD Standard outperforms all compared methods except for the Ground-truth GP and the Ground-truth HGP. It is expected that BO with the ground-truth GP or the ground-truth HGP would outperform all transfer-learning methods as the ground-truth GP and HGP are the actual distributions used to generate the Synthetic Super-dataset and the ground-truth distributions are unknown to the transfer-learning methods. In practice, the ground-truth prior for a real-world super-dataset are unknown and it would be unrealistic to use methods like the Ground-truth GP and the Ground-truth HGP which requires access to the ground-truth prior. Here we include the Ground-truth GP and Ground-truth HGP in Fig. 7 as two non-beatable benchmarks because we know the ground-truth for the Synthetic Super-dataset and with them in the figure we only intend to see how close the proposed method can get to the ground-truth and we do not expect any realistic method that does not have access to the ground-truth prior to surpass the ground-truth. We have also made this clear in the caption of Fig. 7.
>
> **(2/2)**

---

### Review · Reviewer_Y6Xy · 2023-11-13

**Summary Of Contributions:**

Bayesian Optimization is used a lot to optimize black-box functions, e.g., in Hyperparameter Optimization. However, it is not always easy to specify the kernel of a GP since there is a priori only very little knowledge about the black-box function at hand. The authors propose MPHD that meta-learns based on domain-specific contexts how to specify the parameters of a hierarchical GP and thus improve the optimization performance. The big advantage compared to previous approaches is that the search space does not need to be the same on previous functions. They demonstrate the strength of MPHD on artificial functions (with known optimal behavior) and on well-established HPO benchmark (HPO-B and PD1).

**Audience:**

Yes

**Claims And Evidence:**

No

**Requested Changes:**

The paper needs a complete restructuring s.t. it is easy to read and understand.

Furthermore, the results are not convincing in more than one dimension and need improvements (see above).

**Strengths And Weaknesses:**

Strengths:
1. The approach is novel and very well motivated. In fact, it is in practice quite common that previous search spaces are not the same as future search spaces. How to handle this case efficiently is an open research question and the authors provide an interesting step toward solving this problem.
1. The approach is very well mathematically motivated and presented.
1. The benchmarks are extensive and thorough and provide a clear picture of the strength and weaknesses of MPHD and other baselines.
1. Related work is well covered.

Weaknesses:
1. First of all, I got the impression that the authors tried to squeeze the paper into the 12 pages for TMLR. Unfortunately, many very important explanations and details can only be found in the appendix s.t. I had to go forth and back all the time. This makes it very hard to read the paper. Actually, I would strongly prefer to have 16+ page paper here without needing to jump to the appendix again and again.
1. Similarly, I had to jump to the figures forth and back. At the very beginning, the authors refer to Figure 6 which is at the end, and at the end of the paper they refer to Figure 1. Also very annoying.
1. Also the conclusion/discussion at the end is very brief and misses to fully discuss the strengths and weaknesses of the approach and future directions.
1. (to continue with readability): The plots use too many colors without any marking (other than solid and dashed lines). It makes it very hard for me to distinguish the lines and match them with the baselines. People with color blindness would have an even harder time.
1. Very late, I fully understand what the “context” actually is. On Page 9, they say that they use “whether the domain is continuous or discrete, the number of continuous dimensions and the number of discrete dimensions in the domain”. On the one hand, I can understand that there is not much known about a new black-box function; on the other hand, I wonder how much can be learned about this little information. In fact, I wonder whether the entire approach is only working fairly decently because of the hierarchical GP that needs to be adjusted according to these context features.
1. I missed a lot of details on the training of the deep neural networks used for the pre-training.
1. How does the approach compare against a simple hyperparameter tuning of BO’s own hyperparameters on previous functions (e.g., as done by https://ieeexplore.ieee.org/document/9913342) and transferring that to similar functions?
1. Most of the above can be fairly easily improved, but the results are my main concern. Only on the artificial data (which is well designed for the use case), MPHD has a very convincing performance. On the other two HPO benchmarks, MPHD is either only marginally better (maybe not statistically significant) or even worse. Furthermore:
    * In the results, which of the results is a standard BO-GP approach as one would implement it typically (e.g., the old but well-known Spearmint)?
    * Why are baselines missing on PD1?
    * I was surprised that the authors highlighted the results with PI since it is a rather uncommon choice. Also checking the results of Wang and Jegelka 2017, PI is not superior in their results.

---

> ### Author Response · Authors · 2023-11-27
>
> Thank you for your thoughtful feedback. We are glad that you found our approach novel and very well motivated. We are encouraged that you found our theoretical analysis mathematically motivating, our experiments extensive and thorough, and related work well covered. We address your specific comments below and have incorporated your feedback in the revised paper.
>
> > 1. First of all, I got the impression that the authors tried to squeeze the paper into the 12 pages for TMLR. Unfortunately, many very important explanations and details can only be found in the appendix s.t. I had to go forth and back all the time. This makes it very hard to read the paper. Actually, I would strongly prefer to have 16+ page paper here without needing to jump to the appendix again and again.
>
> Thank you for suggesting making the paper longer to incorporate more details in the main paper. We agree that it would be better to put more of the explanations and details in the main paper so that readers would need to jump to the appendix less. We have expanded the paper and included more contents in the main paper: more detailed description of the domain-specific contexts (Section 2), proof of Lemma 1 and intuitions on the theoretical analyses (Section 3.1), details on the NN architecture and training method (Section 3.2.2 and Section 4.2), and additional discussion on the advantages, limitations, and directions for future work (Section 5).
>
> > 2. Similarly, I had to jump to the figures forth and back. At the very beginning, the authors refer to Figure 6 which is at the end, and at the end of the paper they refer to Figure 1.
>
> We referred to Fig. 1 (now Fig. 2) throughout the paper, including Sections 2.1, 2.2, 3.2, 4.2, so we would like to keep its location at where the figure is referred to for the first time. We agree with the reviewer that Fig. 6 (now Fig. 1) needs to be relocated to allow better reading experience, and we have moved it to the beginning of the paper.
>
> > 3. Also the conclusion/discussion at the end is very brief and misses to fully discuss the strengths and weaknesses of the approach and future directions.
>
> Thank you for this suggestion. We discussed the limitation and future direction in Appendix A "Discussions" of our original submission. We have moved those to the main paper, and expanded the discussions in the revised paper to include more details on the strengths and weaknesses of the approach and future directions.
>
> The following is a summary of strengths and weaknesses:
>
> *Strengths: The key advantage of MPHD as a transfer learning approach for BO is that it allows transferring between different search spaces. Moreover, MPHD does not need access to BO trajectories in the format of an ordered list of data points. Instead, MPHD can effectively make use of an unordered set of datapoints as long as they are partitioned to different functions and different domains (see the definition of a super-dataset in section 2.1). As shown in our paper, MPHD enjoys appealing asymptotic properties and performs competitively on challenging hyperparameter tuning tasks, making it both a theoretically sound and a practical transfer learning method.*
>
> *Weaknesses / limitations / future directions: For a BO task, MPHD only learns the prior model in the form of a domain-specific hierarchical GP. While this allows a separation of model and decision making strategy, there are other components that can also be meta-learned, such as acquisition functions, to maximize the effectiveness of the BO system. One direction of future work is jointly pre-train all components of BO to allow more flexibility and further improve the performance.*
>
> *In addition, like most machine learning methods, MPHD is subject to assumptions, including a stationary kernel function, a constant mean function and the availability of a super-dataset with domain-specific contexts. Possible future work includes relaxing  assumptions on kernel and mean functions and incorporating architecture search to enrich the space of hierarchical GP priors. The assumptions on data are naturally satisfied in our experiments such as HPO-B, since our context encodings only require input dimensions and whether the input is continuous. However, for some other types of data, the domain-specific contexts might not be real vectors. Our work builds a strong foundation for generalizing to those more complex contexts if they can be encoded as real vectors, and a future work is to work with those more complex domain-specific contexts.*
>
> **(1/4)**

---

> ### Author Response · Authors · 2023-11-27
>
> > 4. The plots use too many colors without any marking (other than solid and dashed lines). It makes it very hard for me to distinguish the lines and match them with the baselines. People with color blindness would have an even harder time.
>
> Thanks for suggesting updated the colors and styles of the plots to help readers see the performances of different methods better. We have adjusted the colors and introduced additional line styles in Figures 7, 8, 9, 12, 13, 14 in the revised paper and we hope these modifications would help the readers read the results more clearly.
>
> > 5. Very late, I fully understand what the “context” actually is. On Page 9, they say that they use “whether the domain is continuous or discrete, the number of continuous dimensions and the number of discrete dimensions in the domain”. On the one hand, I can understand that there is not much known about a new black-box function; on the other hand, I wonder how much can be learned about this little information. In fact, I wonder whether the entire approach is only working fairly decently because of the hierarchical GP that needs to be adjusted according to these context features.
>
> Please see the general response on domain-specific contexts.
>
> > 6. I missed a lot of details on the training of the deep neural networks used for the pre-training.
>
> We introduced the detailed setup of the compared methods in Appendix E.2, including the kernel and mean functions of the GP, the NN architecture (number of hidden layers, sizes of hidden layers, and activation function), and training method (optimizer, learning rate, and number of training iterations). To improve readability, we have also moved more details of the NN architecture and training method to Sections 3.2.2 and 4.2 in the revised paper given that we are no longer constraining the paper to be within 12 pages.
>
> > 7. How does the approach compare against a simple hyperparameter tuning of BO’s own hyperparameters on previous functions (e.g., as done by https://ieeexplore.ieee.org/document/9913342) and transferring that to similar functions?
>
> We assume that "a simple hyperparameter tuning of BO’s own hyperparameters on previous functions" means transfer learning BO methods on the same search space (otherwise BO's own hyperparameters on previous functions cannot be applied directly to the new function). As shown in Section 4.2, we compared to such kind of methods, including HyperBO (Wang et al., 2022), ABLR (Perrone et al., 2018) and FSBO (Wistuba and Grabocka, 2021). These methods all learn the GP (i.e., tune the GP's hyperparameters) on previous functions and use the learned GP for BO on new functions. They differ in the details of learning and GP model setups. Given that we have a range of other baseline methods and these papers have already shown that they achieve top performance compared to other existing transfer learning BO methods, we did not include other single search space transfer learning BO methods. The IEEE paper that the reviewer pointed out focuses on multi-fidelity hyperparameter optimization, which is not the main focus of our work (how to do transfer learning across heterogeneous domains).
>
> **(2/4)**

---

> ### Author Response · Authors · 2023-11-27
>
> > 8. Most of the above can be fairly easily improved, but the results are my main concern. Only on the artificial data (which is well designed for the use case), MPHD has a very convincing performance. On the other two HPO benchmarks, MPHD is either only marginally better (maybe not statistically significant) or even worse.
>
> We disagree with the interpretation of the results on the HPO-B Super-dataset and PD1 Dataset as negative performances of the proposed method MPHD. It is true that some of the baselines - specifically HyperBO, FSBO, and ABLR - achieve a comparable or better performance than MPHD in experiments on the HPO-B Super-datasets and the PD1 Dataset (NToT). But this is not a negative result for MPHD because these baselines are single-domain transfer learning methods that can only work if there is sufficient training data available from the same domain as the test domain, and these methods need to have a model pre-trained separately for each of the test domains with the corresponding same-domain training data. This is indeed the case for the HPO-B Super-dataset, where these single-domain methods can pre-train their GP parameters successfully for every test domain with access to sufficient same-domain training data. On the other hand, MPHD is a method that works over heterogeneous domains, and can still be used when there is no training data from the test domain but only training data from other different domains. We do not expect that MPHD to outperform these strong single-domain transfer learning methods in the setting where there is single-domain training data available as it is not a fair comparison. MPHD is at a disadvantage when compared with these single-domain methods as (1) MPHD (NToT) is not even pre-trained on the test domain and (2) even when the non-NToT MPHD is pre-trained on the test domain, it pre-trains only one model to be used in all test domains and thus will need to identify the differences of different test domains at test time while the single-domain transfer learning methods explicitly pre-trains a separate model for each of the test domains. **Therefore, we view the fact that MPHD can achieve a comparable performance with these strong single-domain transfer learning methods on a large-scale real-world HPO super-dataset such as HPO-B as a very positive result for MPHD.**
>
> > 8.1. In the results, which of the results is a standard BO-GP approach as one would implement it typically (e.g., the old but well-known Spearmint)?
>
> Out of the compared baselines, the Non-informative HGP and Hand-specified HGP are the closest to a standard BO-GP approach which does not use transfer learning to learn the GP parameters from previous data.
>
> > 8.2. Why are baselines missing on PD1?
>
> Compared to the experiments on the Synthetic Super-dataset and the HPO-B Super-dataset, all the baselines (Random, Base GP, HyperBO, ABLR, FSBO, Hand-specified HGP, and Non-informative HGP) are all included in the experiments on PD1 Dataset. Ground-truth GP and Ground-truth HGP are not included in PD1 experiments just because there is no known ground-truth GP or HGP for the PD1 Dataset. The only remaining difference is that the non-NToT versions of MPHD Standard and MPHD Non-NN are not included in the PD1 experiments, and this is because the PD1 experiments are used to emphasize the generalization capability of MPHD to a domain that is unseen during pre-training.
>
> In our experiments we use the Synthetic Super-dataset and HPO-B Super-dataset as the two super-datasets to do comprehensive comparisons between all versions of MPHD and baselines. We use PD1 Dataset as an extra test dataset for the MPHD model pre-trained on HPO-B to demonstrate the capability of MPHD to generalize its model pretrained on one real-world super-dataset (HPO-B) to a different real-world dataset (PD1) that is unseen during pre-training. With this specific emphasize on transferring across different real-world datasets (and therefore also domains) as the purpose of the PD1 experiments, we include the MPHD version that is only pre-trained on HPO-B but not on PD1 at all [MPHD Standard (NToT)] to compare with baselines. We did not include MPHD Standard (which would be a MPHD model pre-trained on both HPO-B and PD1) in the PD1 experiments because it does not fit in the setting of generalizing to an unseen domain that we want to showcase with PD1, and comparisons of MPHD Standard and MPHD Standard (NToT) are already included in experiments on the two super-datasets. For the similar reason, MPHD Non-NN (NToT) is included in the PD1 experiments while MPHD Non-NN HGP is not.
>
> **(3/4)**

---

> ### Author Response · Authors · 2023-11-27
>
> > 8.3. I was surprised that the authors highlighted the results with PI since it is a rather uncommon choice. Also checking the results of Wang and Jegelka 2017, PI is not superior in their results.
>
> Wang and Jegelka (2017) proposed the max-value entropy search method, a special case of which is equivalent to a version of PI with target value set to be an estimate of the function maximum value (see their Lemma 3.1). Figure 1 and Table 1 in Wang and Jegelka (2017) show that this special case (MES-R 1 and MES-G 1) achieves competitive results to the generic version of their proposed method. In their experiments, "PI" corresponds to a classic version of PI in Kushner, 1964, as shown in their citation of PI. On the contrary, the version of PI in our paper uses a rough estimate of the function maximum value, in a similar fashion to the special case investigated by Wang and Jegelka (2017). We also conducted experiments using GP-UCB and EI, and the results are shown in Fig. 12, 13 and 14. We found that PI (the version in our paper) gives either comparable or slightly better results than the other two methods, so we included PI results in the main paper. To ensure the main paper is clear, we put the GP-UCB and EI results in the appendix. We have clarified these points in the main paper.
>
> > Requested change: The paper needs a complete restructuring s.t. it is easy to read and understand.
>
> Thanks for your suggestions. We have made modifications in multiple aspects in the revised paper to improve readability. Please see our responses to 1, 2, 3, 4, 5, 6 and the general response.
>
> > Requested change: the results are not convincing in more than one dimension and need improvements.
>
> Please see our responses to your specific comments about the experiment results above (8, 8.1, 8.2, 8.3).
>
> **(4/4)**

---

### Author Response · Authors · 2023-11-27
**General response to reviewers' comments**

We thank the reviewers for their thoughtful feedback.  We are encouraged that they found our method intuitive, natural (Reviewer YACK), interesting, reasonable (Reviewer MKMS), novel, and well motivated (Reviewer Y6Xy). We are glad that they found our theoretical analysis interesting (Reviewer YACK), mathematically motivating (Reviewer Y6Xy), and found our experiments competitive (Reviewer YACK), extensive, and thorough (Reviewer Y6Xy). We are pleased that Reviewer MKMS recognizes the value and importance of our problem setting and that Reviewer Y6Xy recognizes related work being well covered in the paper.

## Common question about domain-specific contexts
All reviewers recommended adding a more detailed description of the domain-specific contexts and/or asked questions about the context vector. Therefore we address this common question here.

We illustrated examples of context configurations for length-scales in Fig. 2 (now Fig. 3), which correspond to "whether the **domain dimension** is continuous or discrete, the number of continuous dimensions, and the number of discrete dimensions in the domain" in Section 4.2. There was also a more detailed description of the contexts in Appendix F.2 (now Appendix E.2). However, we agree that an even more detailed elaboration on how the contexts are constructed would be helpful for the readers to understand it better and therefore we have added more detailed descriptions and examples of contexts in Section 2.1 of the revised paper.

Note that in Section 4.2, we used a 4-dimensional context for each length-scale parameter in the kernel. For a d-dimensional domain using Matern kernel, there are d length-scales and so there are 4*d dimensional contexts in total, which we consider to carry quite a bit of information for a black-box function. In both the synthetic and real-world experiments (Figures 7, 8, 9, 12, 13, 14), we conducted ablation studies where we compare MPHD with the contexts (MPHD Standard) and MPHD without the context information (MPHD Non-NN), and our results show that MPHD Standard significantly outperforms MPHD Non-NN. This proves that the contexts do carry a descent amount of information about the function and our model is able to effectively learn from this information. Adjusting the hierarchical GP according to contexts is non-trivial, and MPHD provides a principled approach to do so. We have revised and made these points clearer.

There could be other ways to include more sophisticated information about the black-box function, and MPHD is a general method to incorporate any kinds of contexts in a vector format. Investigating those scenarios would require a considerable amount of work to create new benchmarks with more complex contexts, and we plan to pursue these directions in the future.

## Additional comments and main changes in the revised paper

**We address additional specific comments by responding to each reviewer separately. In addition, we would like to highlight here the main changes we have made in the revised paper in response to common comments from the reviewers.**

* Added more detailed descriptions and examples of the domain-specific contexts in Section 2.1 to help readers understand it better.

* Added more detailed and rigorous proofs of the theoretical results (Theorems 2 and 3) in Appendix D. Added the proof of Lemma 1 and intuitions on the theoretical analyses to the Section 3.1 to allow better readability.

* Moved more details of the NN architecture and training method to the main paper (Sections 3.2.2 and 4.2).

* Adjusted the colors and added more line styles in the BO experiment figures (Figures 7, 8, 9, 12, 13, 14) to make it easier to read.

* Expanded the conclusion section to include more discussions of the advantages, limitations of MPHD, and directions of future work.

---

### Decision · Action_Editor_vczy · 2024-01-10

**Recommendation:** Accept with minor revision

**Comment:**

The reviewers value the studied problem and the contributions made, as well as the empirical results. The majority agrees that the paper would be suitable for publication at TMLR.

However, all reviewers flagged in their reviews that the notion of context was not particularly clear. While the majority of the reviewers agree that the revision has alleviated that issue, and resolved many other, they do remain sceptical about the amount of information in the context features used in the simulations.

   In the minor revision, hence please

   1. add good intuitive arguments to the paper on why and how the simple context information provides the information needed for the method to work as intended.

   When reading the paper, I was confused by the following two additional points:

   2. Fig 1, right column (test domain T): why does the green box indicate that the length-scales / noise variances of domain T are inputs for the learned model \hat{phi} alongside the context? My understanding from Section 2.2 and Eq 6 is that the model phi only takes the context as input, and that parameters such as the length scales are thought to be samples from the prior with hyperparams phi(context), with their MAP estimates being used for BO on a test function (eq 6). Please clarify or amend Fig 1.

   3. Sec 3.2 (synthetic super-dataset L): the hyperparameters are said to depend linearly on the dimensions of each domain. Please brielfy explain the motivation for this, e.g. what functional properties are you modelling?

**Audience:**

The paper studies transfer learning for Bayesian optimisation where domains may different properties (e.g. some may be discrete while others are continuous). The topic will be of interest to a wider readership of TMLR.

**Claims And Evidence:**

The claims are supported both by theory or simulation results.

---

> ### Author Response · Authors · 2024-02-01
>
> We thank the action editor and the reviewers once more for their insightful comments. We are pleased with the decision and have made some minor updates to the camera-ready version of the paper, addressing the specific points raised by the action editor.
>
> > Add good intuitive arguments to the paper on why and how the simple context information provides the information needed for the method to work as intended.
>
> Intuitively, the prior over a continuous learning rate hyperparameter should be very different from the prior over a discrete hyperparameter which refers to an activation function type. Moreover, tuning problems on similar models tend to have a similar number of hyperparameters to tune. For example, in HPO-B (Pineda-Arango et al., 2021), the "svm" search spaces have 6 to 7 dimensions, while the "rpart" search spaces have 29 to 31 dimensions. We have incorporated this intuitive explanation in Section 2.1.
>
> > Fig 1, right column (test domain T): why does the green box indicate that the length-scales / noise variances of domain T are inputs for the learned model $\hat{phi}$ alongside the context? My understanding from Section 2.2 and Eq 6 is that the model phi only takes the context as input, and that parameters such as the length scales are thought to be samples from the prior with hyperparams phi(context), with their MAP estimates being used for BO on a test function (eq 6). Please clarify or amend Fig 1.
>
> The understanding of Section 2.2 and Eq. 6 is correct that the model $\hat{\phi}$ only takes the context of a GP parameter (such as the length-scale of a specific domain dimension) as input, but not the GP parameter itself. The right column of the earlier version of Fig. 1 is also intended to demonstrate the same pipeline. The text in the green box, "$S_{T1}$: context of $\theta_{T1}$, length-scale of 1st dimension of domain T", is meant to show that the context $S_{T1}$ of the length-scale GP parameter is the only input to model $\hat{\phi}$, and that $\theta_{T1}$ is the notation of the GP parameter but not part of the input. We agree that this text can be confusing to readers, and have changed the wording to "$S_{T1}$: context of length-scale $\theta_{T1}$ of 1st dimension of domain T" in the camera-ready version.
>
> > Sec 3.2 (synthetic super-dataset L): the hyperparameters are said to depend linearly on the dimensions of each domain. Please brielfy explain the motivation for this, e.g. what functional properties are you modelling?
>
> We let the hyperparameters depend on the number of input dimensions in order to simulate practical applications where tuning objectives with different numbers of input dimensions might need to be modeled with very different hyperparameters in MPHD. We further set this dependency to be linear in the synthetic experiments, so that we can compare the learned hyperparameters against the actual ground truth. The choice of using a linear dependence is not to model any particular functional properties, but rather because it is a simple functional form that is easy to understand. We have added brief explanation for this in Section 3.2.